# Quantitative rainfall analysis of the 2021 mid-July flood event in Belgium

Michel Journée[1], Edouard Goudenhoofdt[1], Stéphane Vannitsem[1], and Laurent Delobbe[1]

[1]Royal Meteorological Institute of Belgium, Avenue Circulaire 3, 1180 Brussels, Belgium

**Correspondence:** Michel Journée (michel.journee@meteo.be)

**Abstract.** The exceptional flood of July 2021 in central Europe impacted Belgium severely. As rainfall was the triggering factor of this event, this study aims at characterizing rainfall amounts in Belgium from July 13th to July 16th 2021 based on 2 types of observational data. First, observations recorded by high-quality rain gauges operated by weather and hydrological services in Belgium have been compiled and quality checked. Second, a radar-based rainfall product has been improved to provide a reliable estimation of quantitative precipitation at high spatial and temporal resolutions over Belgium. Several analyses of these data are here performed to describe the spatial and temporal distribution of rainfall during the event. These analyses indicate that the rainfall accumulations during the event have reached unprecedented levels over large areas. Accumulations over durations from 1 to 3 days significantly exceeded the 200-year return level at several places, with up to 90% of exceedance over the 200-year return level for 2-day and 3-day values locally in the Vesdre basin. Such a record-breaking event needs to be documented as much as possible and available observational data must be shared with the scientific community for further studies in hydrology, in urban planning and, more generally, in all multi-disciplinary studies aiming at identifying and understanding factors leading to such disaster. The corresponding rainfall data are therefore provided freely as a supplement (Journée et al., 2023; Goudenhoofdt et al., 2023).

## 1   Introduction

From July 13 to 16 2021, a long period of sustained and heavy rainfall affected Central Europe producing extreme rainfall amounts in western Germany, eastern Belgium, Luxembourg and The Netherlands. This meteorological event combined with already near-saturation soil conditions and steep slopes of several river valleys in the region caused disastrous flooding (Kreienkamp et al., 2021; Mohr et al., 2022). This event was one of the most severe natural catastrophes in Europe in the last half century and was responsible for at least 220 fatalities and loss amounts estimated to EUR 46 billion (MunichRe, 2022; Mohr et al., 2022). In the Walloon region of Belgium alone, 39 people lost their lives and the total economic damage for this region is estimated to EUR 2.8 billion (Gouvernement Wallon, 2022). According to Assuralia (2022), the total amount of compensations paid in Belgium by the insurance companies is EUR 2 billion. This exceptional event occurred within a low-pressure system, called Bernd, very slowly moving over Central Europe, feeding the flooded region with very moist air. The associated rainfall was characterized by a sustained large-scale stratiform component combined with locally embedded convective precipitation. In eastern Belgium, this led to extreme rainfall amounts breaking many historical rainfall records at several

locations. Exceptional floods were observed in several tributaries of the Meuse catchment, in particular along the Vesdre (see Figure 1) where the most severe consequences have been deplored in terms of casualties and damage (Dewals et al., 2021). In the Vesdre catchment, an increase of the seismic noise and an increasing saturation of the weathered zone have been observed during this event by a seismometer and a gravimeter, respectively (Van Camp et al., 2022). The increase of the seismic noise

during the event is induced by the rising stream turbulence, sediment and debris transport. On 15 July at 00:15 UTC a sudden increase of the seismic noise coincides with the detailed testimony reporting a sudden roaring in the valley before the arrival of a flash flood, described as a "tsunami", 3 km downstream of the geophysical station. The gravimeter is installed 48m underneath the surface and signal variations are induced by water accumulating above the gravimeter. The evolution of the gravity measurements along the event shows increasing subsoil's saturation with less and less water accumulation and increasing run

off.

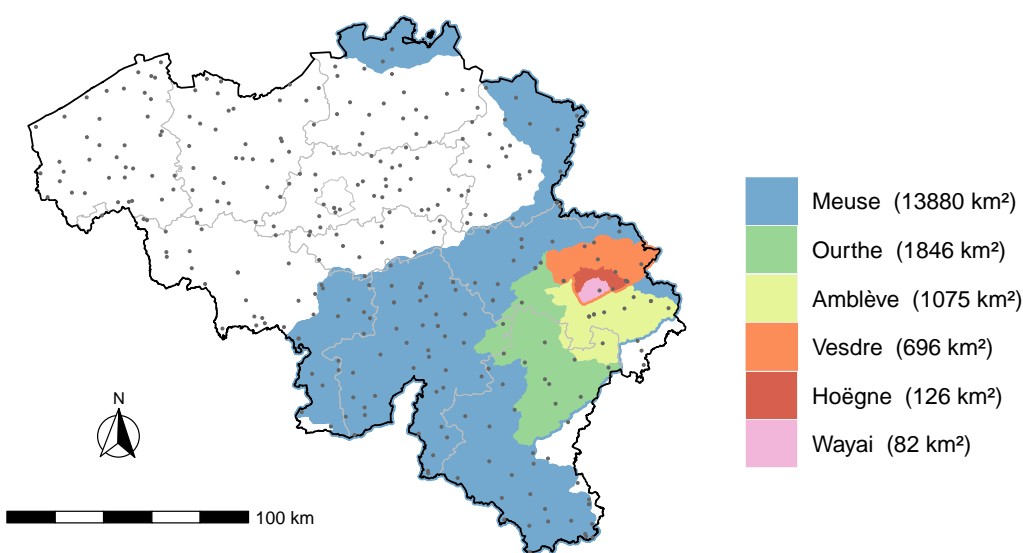

**Figure 1.** Map of Belgium with the definition of the 6 catchments discussed in this study. Some of these domains are overlapping: the Wayai and Hoëgne catchments are included in the Vesdre catchment; the Vesdre, Ourthe and Amblève catchments are included in the Meuse catchment. The Meuse catchment is here limited to its Belgian part. The dots represent the locations of the rain gauges that provided data during the 2021 mid-July event. The gray lines delimit the Belgian provinces.

After such a disaster, questions arise on the role of climate change on the occurrence of this type of event. An attribution study has been rapidly performed (Kreienkamp et al., 2021) and concludes that the likelihood of such an event to occur today at any place over Western Europe compared to a 1.2 °C cooler climate has increased by a factor between 1.2 and 9. According to the Clausius-Clapeyron relation, a warmer atmosphere can contain more water vapour which is expected to increase the

intensity and frequency of precipitation events. This type of relationship is however difficult to ascertain as the occurrence of

such an event is both dynamically and thermodynamically driven (Ludwig et al., 2022; IPCC, 2021; Vergara-Temprado et al., 2021). This makes the attribution of such events to climate change a real challenge, and further analyses are required to address this issue as for instance in Meyer et al. (2022). Such type of analyses strongly rely on the quality of the available observational data.

As described in Mohr et al. (2022), these devastating floods are the result of complex interactions between meteorological, hydrological and hydro-morphological phenomena. An additional aspect that needs to be considered is the presence of dams upstream of some affected valleys in eastern Belgium (Dewals et al., 2021). Multi-disciplinary analysis are required for an in-depth understanding of the course of the events. Investigating the complex dynamics in the affected catchments is necessary to understand the relation between extreme rainfall and the resulting impact. Such analyses require a detailed knowledge of

rainfall, as the triggering factor of these events. Extremely rare events need to be documented as much as possible and data must be made available for further studies in hydrology, in urban planning and, more generally, in all multi-disciplinary studies aiming at identifying and understanding factors leading to such disaster.

Several extreme rainfall and flood events that occurred in the last decades have been documented in the literature. For example, the extraordinary rainfall and flash flood event which affected the eastern part of The Netherlands in August 2010 is

described in Brauer et al. (2011). In Germany, the hydrometeorology of the extreme flood in the Starzel catchment on 2 June 2008 is analysed in Ruiz-Villanueva et al. (2012). On 8-9 September 2002, a catastrophic flash flood affected the Gard region (France) with maximum 24h rainfall values of 600-700 mm. This event is extensively documented and analysed in Delrieu et al. (2005). In Marchi et al. (2010), hydrometeorological data from 25 extreme flash floods across Europe are presented and a high-resolution dataset of rainfall and discharge observations for 49 events in Europe and the Mediterranean region from

1991 to 2015 is described in Amponsah et al. (2018). All these studies underline the importance of rainfall observations at high spatial and temporal resolutions for the post-analysis of extraordinary flood events.

In the present paper, the observational rainfall data for Belgium that can be used for such analyses are exposed and made available to the scientific community (Journée et al., 2023; Goudenhoofdt et al., 2023). These data are twofold: (1) in-situ observations from high-quality rain gauges and (2) rainfall products based on weather radar observations. The radar data are

carefully processed and combined with rain gauge measurements to provide a quantitative estimation of precipitation at high spatial (i.e., 1 km) and temporal (i.e., 5 min and hourly) resolutions. Several analyses of these data are performed to describe the spatial and temporal distribution of rainfall during the event and to illustrate its exceptional character.

The paper is organized as follows. In Section 2, the rain gauge data and the radar product available for Belgium during the period from July 13th to July 16th 2021 are detailed. Section 3 includes several analyses of these data to discuss the event

regarding its spatial and temporal extent together with its exceptional character. Some conclusions are drawn in Section 4.

## 2 Precipitation data

### 2.1 Rain gauge data

Several rain gauge networks are deployed in Belgium to continuously monitor rainfall at ground level. These networks are operated by the Royal Meteorological Institute of Belgium (RMI) and by the Belgian regional hydrological services, i.e., Service Public de Wallonie - Mobilité et Infrastructures (https://hydrometrie.wallonie.be), Vlaamse Milieumaatschappij and Hydrologisch Informatiecentrum (https://www.waterinfo.be/). In total, 323 rain gauges located in 308 different sites have recorded precipitation quantities during the severe rainfall event of mid-July 2021. The spatial distribution of these 308 sites is illustrated in Figure 1 by the grey dots. On average over Belgium, this represents a density of 10 measurement sites per 1000 $km^2$. This density is somewhat lower for the Meuse basin (9 per 1000 $km^2$) and slightly larger for the Vesdre basin (11.5 per 1000 $km^2$). Among the 323 available devices, 168 weighing rain gauges monitored rainfall in real-time with a 5-min time resolution and 155 manual rain gauges recorded daily precipitation amounts. Manual precipitation observations are made each day at 08:00 a.m. local time.

All the recorded rain gauge data were checked manually for possible errors and inconsistencies. This data quality control (QC) analysis is made with the help of maps and time series plots that allow to compare data from neighboring rain gauges. Comparisons are also made with meteorological radar products. This validation process highlighted that 2 weighing rain gauges provided zero precipitation data continuously during several hours, which was inconsistent with the heavy rainfall observed by neighboring rain gauges as well as by the radars during that period. In addition, a few gaps of short duration (usually 1 or 2 successive timestamps) were reported in the 5-min time series of some other weighing rain gauges. Estimations derived by inverse distance weighted interpolation (IDW) from neighboring weighing rain gauge data have been considered to correct the 2 periods of erroneous data as well as to fill the gaps. Regarding the manual rain gauges, some observers measured the daily precipitation amount later than 08:00 am. As a consequence, the total accumulation during the event was correctly recorded but not properly distributed on each day, needing adjustments of the daily values. This adjustment is based on the sequence of daily totals of neighboring rain gauges. Globally, the QC analysis led to few interventions on all available rain gauge data and none concerning the rain gauges located in the most severely affected area (i.e., Vesdre basin).

Finally, one should note that rain gauge data are subject to various sources of uncertainties. First, the World Meteorological Organization (WMO) recommends the measurement uncertainty related to the sensor performance under nominal and recommended exposure to stay below 5% (WMO, 2018a). Second, the close environment of the rain gauge might influence the measurements. Following the WMO siting classification (WMO, 2018b), RMI weighing rain gauges are classified either as Class 1 (i.e., reference site) or Class 2 (i.e., additional estimated uncertainty added by siting up to 5%) (Gonzalez Sotelino et al., 2016). No classification is currently available for the other rain gauges considered in this analysis.

### 2.2 Radar-based rainfall product (RADFLOOD21)

The radar product is obtained after a careful processing of the weather radar measurements and a merging with validated rain gauge measurements. Some of the main challenges of radar-based precipitation estimation are discussed in detail in

Goudenhoofdt and Delobbe (2016). The method is under a continuous improvement process based on research and quality
control. In particular, the significant underestimation of the operational product during the flood event, has led to several
improvements. Additionally, some of the parameters have been optimized for this particular case. The processing steps are
explained below.

### 2.2.1   Weather radar measurements

Radars transmit electromagnetic pulses, typically within a beam width of 1 degree. Part of the transmitted power is reflected
back to the radar by precipitation. Most radars in Europe perform a full volume scan of the atmosphere at different elevation
angles (3D) in about 5 minutes. Estimating rainfall from radar measurements is a challenge because of the many sources of
errors and uncertainties (Figure 2).

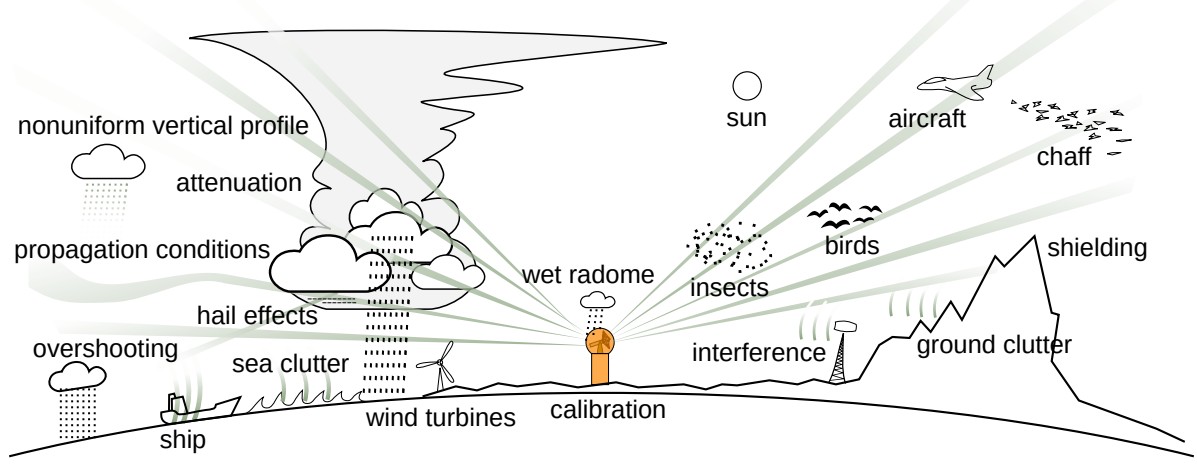

**Figure 2.** Phenomena affecting the radar data quality. Courtesy of Markus Peura (Finnish Meteorological Institute).

The product is based on the 3D reflectivity measurements of the following radars:

–   Helchteren, Vlaamse Milieumaatschappij, Belgium (BEHEL)

–   Jabbeke, Royal Meteorological Institute of Belgium, Belgium (BEJAB)

–   Wideumont, Royal Meteorological Institute of Belgium, Belgium (BEWID)

–   Neuheilenbach, Deutsche Wetterdienst, Germany (DENHB)

–   Essen, Deutsche Wetterdienst, Germany (DEESS)

–   Avesnois, Meteo France, France (FRAVE)

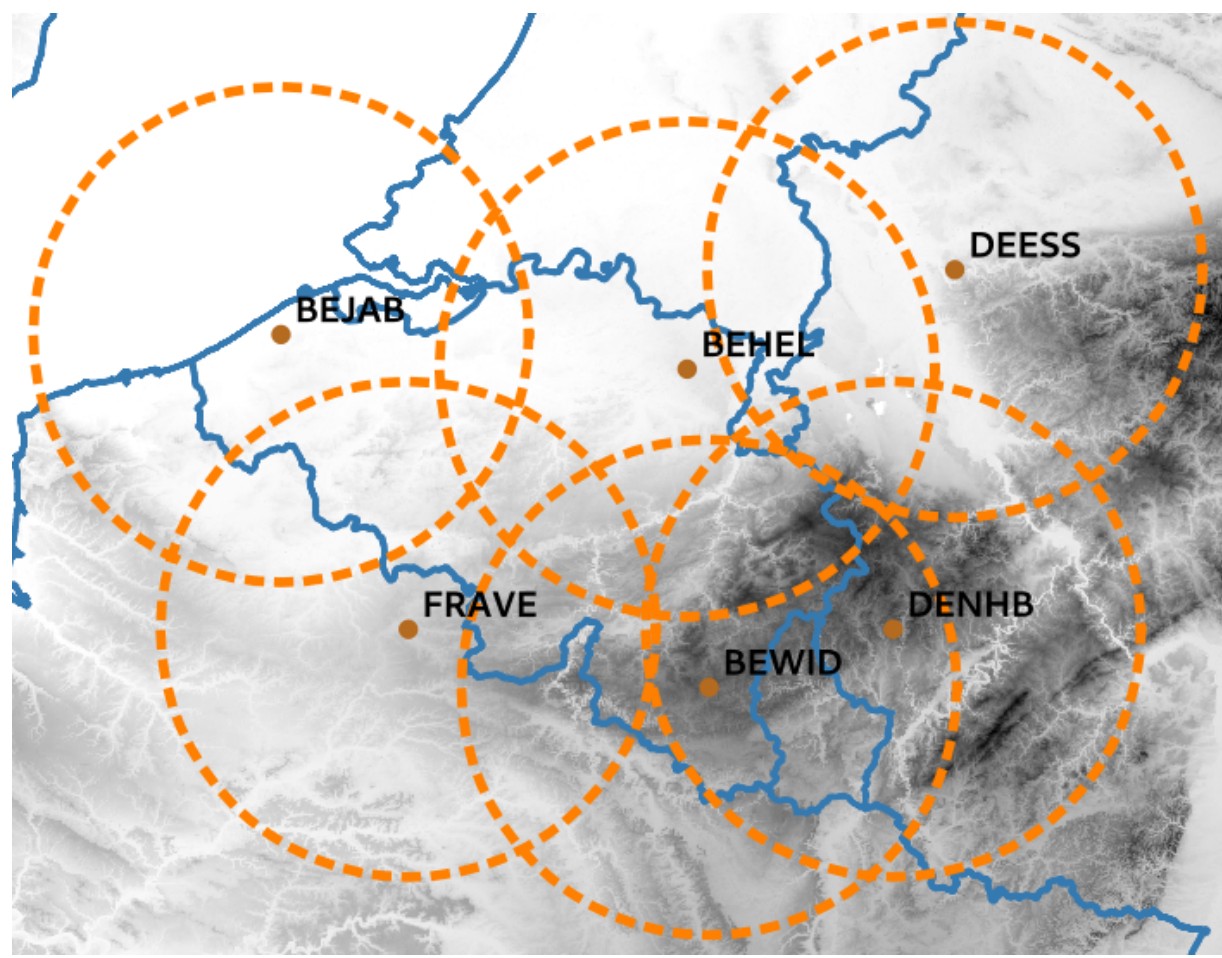

**Figure 3.** Radar coverage with 100 km radius, country borders and height above see level (in gray scale from 0 to 1000 meter).

The radars are all C-band dual-polarization radars except the Wideumont radar which was still a single-polarization radar in 2021. Their configuration and data processing can differ significantly. This leads to inhomogeneities between the radars that need to be reduced before producing a composite.

A radar has a typical range of 250 km but the quality of the measurements tends to decrease with the distance. As found by Goudenhoofdt and Delobbe (2016), data within a 100 km radius are generally considered as appropriate for rainfall estimates of good quality (Figure 3).

### 2.2.2 Quality control of the radar measurements

The quality control starts by checking the long-term calibration bias of the radar. This uses a basic radar estimation method, with interpolated reflectivity at 800m above the radar level converted to rain rate using the Marshall-Palmer relationship ($Z = 200R^{1.6}$). An average bias is computed based on comparison with gauge measurements (collocated pixel) over the past two

months. The method is tuned to remove all uncertainties and errors not related to calibration errors. The calibration bias is removed before further processing.

Radar measurements over a given area can be permanently or regularly affected by clutter (i.e. non-meteorological echoes) coming from hills or wind farms. This can be solved by removing measurements with an abnormally high frequency of echoes. A new method has been developed to avoid filtering these measurements if their values are significantly larger than the maximum expected clutter level. The goal is to include radar observations as close to the ground as possible without contamination by ground echoes. This is important to mitigate underestimation due to orographic enhancement of precipitation taking place in the lowest layers of the atmosphere. We suspect that this effect was particularly strong during the flood event. The new method is presented in details in the Appendix A.

The radar beam can be partially blocked by elevated areas. The percentage of energy lost is therefore computed. The reflectivity measurements is then corrected accordingly.

The radar measurements can be contaminated by residual non meteorological echo from planes, wireless devices or the ground in case of abnormal propagation. Such clutter are identified based on three automatic methods: (1) comparison with the satellite cloudiness product, (2) detection of abnormal changes between measurements at different altitudes (see Appendix B for more details) and (3) detection of unrealistic spatial texture.

### 2.2.3 Rainfall rate estimation at the ground

The processing starts by the identification of convective precipitation based on reflectivity gradients. Non convective precipitation is extrapolated to the ground by using an averaged vertical profile of reflectivity. This allows in particular to mitigate the overestimation caused by melting snow. Missing data after the quality control can be replaced by data from higher radar beams or by data in a close neighborhood. The radar reflectivity ($Z$) is then converted into rain rates ($R$) taking into account the precipitation type. Only rain has been observed for the current event. The current event was characterized by low to moderately high reflectivity.

The standard Marshall-Palmer (MP) relation is normally applied. Above 40 dBZ, the DWD convective relation is used ($Z = 77R^{1.9}$), resulting in lower rain rates with respect to MP. For localized precipitation below 40 dBZ, the US convective relation is used ($Z = 300R^{1.4}$). It also results in lower rain rates compared to MP. A special relation is used for reflectivity below 40 dBZ in areas with orography, where some precipitation enhancement is expected. The NOAA recommended relation for orography in West US is used ($Z = 75R^{2.0}$). This results in higher rain rates compared to MP.

### 2.2.4 Compositing and rainfall accumulation

In order to mitigate the underestimation produced by radar signal attenuation by rainfall along the path, the observations from several radars are combined. Only data from the 3 closest radars and within a range of 180 km are used here. Data at very long range with low spatial resolution are not included.

Instantaneous rain rates are obtained every 5 minutes corresponding to the full 3D radar scan. The rainfall accumulation over 5 minutes is obtained by computing the movement of precipitation using optical flow techniques. The combined local-

global method is used. Note that the present flood case was not characterized by strong winds. Temporal sampling effects (e.g. jumping cell or ripple effects on the accumulation maps) were therefore very limited. Accumulations over longer durations are obtained by taking the sum of the 5-min accumulations.

### 2.2.5 Merging with rain gauges

The accumulated rainfall over 1 hour is combined with rain gauge measurements from the automatic stations using Kriging with external drift (KED) (e.g. Hudson and Wackernagel, 1994). The KED method interpolates the gauge values in a given neighborhood while taking the radar estimation as a linear combination of the expected value of the (Gaussian) process. The method has been tuned for Belgium (see Appendix C). The sliding 1-hour spatial correction factor is applied to the 5 minutes accumulation. This favors the homogeneity of the 5-min products over the consistency with the 1-hour products.

## 3 Quantitative rainfall analysis

### 3.1 Total rainfall distribution in space

Precipitation accumulations over 3 days, from July 13th 06:00 UTC to July 16th 06:00 UTC, vary largely over Belgium, as illustrated on Figure 4. While the weather was completely dry during this period in the North-West of Belgium, rainfall accumulations reached almost 300mm in the eastern part of the country (i.e, the rain gauges recorded up to 291.6mm). Accumulations over the 3-day period exceeding 200mm have been recorded by 4 weighing rain gauges (Jalhay, Spa, Mont Rigi and Neu-Hattlich) and by one manual rain gauge (Hockai). These 5 rain gauges are located in the Province of Liège.

In Figure 4, the 3-day precipitation total is mapped over Belgium in two ways. On one hand, all available 3-day totals recorded by weighing and manual rain gauges are spatially interpolated by an inverse distance weighted (IDW) approach (Figure 4, panel a.). On the other hand, the hourly radar product data are temporally aggregated on the 3-day period (Figure 4, panel b.). Both estimations provide on a large-scale perspective comparable rainfall distributions over the country. Differences are however noticeable at finer scales. First, the IDW approach tends to give artificial circular patterns around some rain gauges, while radar observations are expected to better capture fine-scale rainfall patterns. Second, the IDW approach is blind in areas with a poor density in rain gauges. The difference between both estimations (Figure 4, panel c.) shows that the largest discrepancies are located in an area without any rain gauge in the North of Belgium. Significant differences can also be noted in eastern part of the Province of Liège where the precipitation accumulations are the largest and vary strongly on short distances. Such rainfall patterns with large gradients need a very dense network of rain gauges to be accurately captured solely by ground observations. The radar product is thus essential for the spatial analysis of such precipitation events. In particular, the radar product shows that the 3-day precipitation total has exceeded 200mm for a narrow but elongated area of 265 km$^2$ oriented from southwest to northeast.

In order to estimate the uncertainty associated to both spatial distributions of the 3-day precipitation accumulation, a validation analysis is conducted as follows. As the radar product do not integrate observations from manual rain gauges, the 3-day

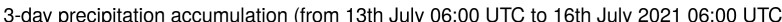

**Figure 4.** Spatial distribution in Belgium (top panels) and in the Province of Liège (bottom panels) of the precipitation accumulation over 3 days (from July 13th 06:00 UTC to July 16th 06:00 UTC). The spatial distribution is derived by inverse distance weighted (IDW) interpolation of all available automatic and manual rain gauge data (left panels) as well as by the radar product (center panels). The difference between both estimations is provided in the right panels. Rain gauge locations are displayed by the grey dots and the red delimited area corresponds to the Province of Liège.

totals from 155 manual rain gauges serve as validation data. For the IDW-derived spatial distribution that relies on both weighing and manual rain gauge data, a leave-one-out cross-validation is applied, i.e., 155 IDW estimations are computed by leaving systematically one different manual rain gauge data out which is used as reference to evaluate the estimation error. The results are illustrated by scatter plots in Figure 5. Concerning IDW, these results indicate an overestimation of the smallest values and an underestimation of the largest ones. IDW tends to smooth the spatial distribution of rainfall and attenuates extremes. Regarding the radar product, a rather good match with the manual observations can be noticed for 3-day accumulations till 100mm. The largest values are however generally underestimated. The derived Mean Absolute Error (MAE) and Root Mean Square Error (RMSE) indicate that the radar product provide nevertheless a more accurate spatial distribution of the 3-day accumulation than IDW.

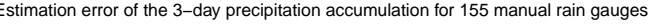

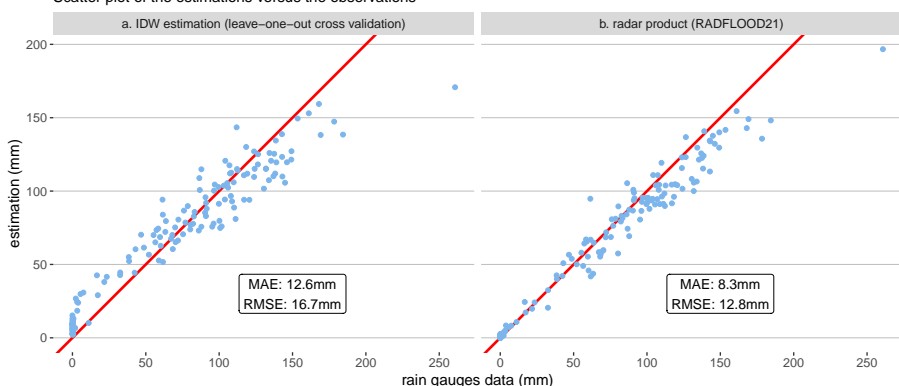

**Figure 5.** Scatter plots of the IDW and radar product estimations versus the observations of the 3-day accumulation for 155 manual rain gauges. The average discrepancies between estimations and observations are summarized by the Mean Absolute Error (MAE) and the Root Mean Square Error (RMSE) for both scatter plots.

By taking a closer look on the rightmost point of the scatter plots of Figure 5 (i.e., manual rain gauge with the largest 3-day accumulation), we find an underestimation of almost 25% by the radar product, while the IDW interpolation of neighboring rain gauges underestimates that specific observation by 35%. A similar analysis but focused on daily rainfall totals also highlights that the estimation of the largest values is challenging but better addressed by the radar product. Daily precipitation values above 80mm, which have been recorded 17 times by manual rain gauges during the 3-day period, are on average underestimated by 16% by the radar product and by 22% by the IDW interpolation of neighboring rain gauges.

In order to assess the exceptionality of the precipitation amounts during the event, extreme precipitation statistics derived for Belgium by Van de Vyver (2012) and Van de Vyver (2013) have been considered. These statistics result from a spatial generalized extreme value (GEV) distribution fitted on annual rainfall maxima derived for various accumulation durations and several locations in Belgium from historical precipitation data. Thanks to this spatial approach, extreme precipitation statistics are estimated for any location in Belgium and are currently considered as the reference for, e.g., the design of hydraulic structures. Return periods can be associated from these statistics to the 3-day precipitation accumulations, as illustrated in Figure 6 for the rainfall distribution estimated by the radar product (i.e., Figure 4b). Large return periods can be noted for extended areas within Belgium. For instance, the return period estimated for the 3-day total exceeds 50 years for 24.5% of the Belgian territory (i.e., 7500 km$^2$). The area with a return period estimated larger than 100 years covers more than 13% of the country (i.e., 4000 km$^2$). It should be noted that the extreme precipitation statistics provided by Van de Vyver (2012) and Van de Vyver (2013) are here limited to return periods of 200 years, as the uncertainty increases considerably when considering return periods significantly larger than the length of the time series. This upper level is exceeded for an area corresponding to 6.5% of Belgium (i.e., 2000 km$^2$). At some places, the 3-day total exceeds considerably the 200-year return level.

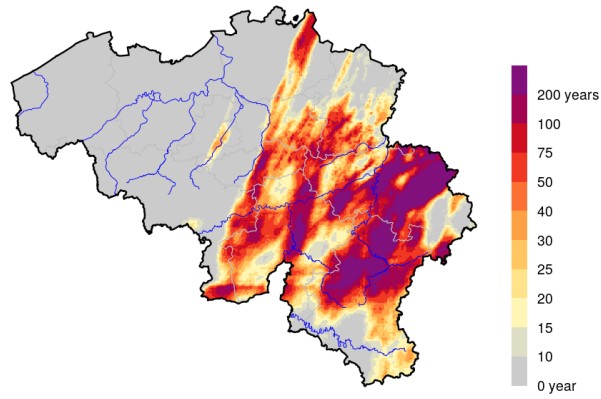

**Figure 6.** Return period estimated for the 3-day precipitation total in Belgium (corresponding to the rainfall distribution estimated by the radar product, i.e., Figure 4b).

## 3.2 Rainfall distribution in time

Figure 7 provides a closer look to the rainfall distribution in time for the 4 weighing rain gauges with a 3-day accumulation over 200mm. These time series indicate that, for these locations, most of the total rainfall accumulation occurred in a period of
225 approximately 36 hours, thus much smaller than 3 days. It appears also that, within this 36-hour period, the hourly precipitation totals are highly variable with several peaks and some hours with almost no precipitation.

For the 5 manual and weighing rain gauges with the largest 3-day total (i.e., exceeding 200mm), the maximum precipitation accumulation for durations from 1 hour to 3 days is provided in Figure 8 and compared against precipitation amounts corresponding to events with return periods from 10 to 200 years, i.e., 10-year to 200-year return levels. For these stations which are
230 all located in the same area of the Province of Liège, precipitation accumulations on short durations till 3 hours do not exceed a return period of 30 years. The precipitation quantities tend however toward extreme levels when increasing the accumulation duration: 6-hour accumulations present a return period above 75 years at 3 places and 12-hour accumulations exceed the 200-year return level at the 4 weighing rain gauges. For the 5 considered locations, the exceedance over the 200-year return level is even larger for precipitation accumulations over 1 day and especially over 2 and 3 days: these 2-day and 3-day values exceed
the 200-year return level by 30% to 90% depending on the location. In Jalhay, where the largest 2-day and 3-day amounts have been recorded, these 2-day and 3-day values correspond to the 200-year return level of rainfall events with a duration of 13 days and 15 days respectively, as it can be noted in Dewals et al. (2021).

The analysis of Figure 8 can be extended to any location in Belgium based on the hourly radar product. For each 1-km pixel of the radar product, the maximum rainfall accumulation is derived for durations from 1 hour to 3 days during the period
July 13th 06:00 UTC to July 16th 06:00 UTC. Return periods can then be associated to these precipitation maxima based on

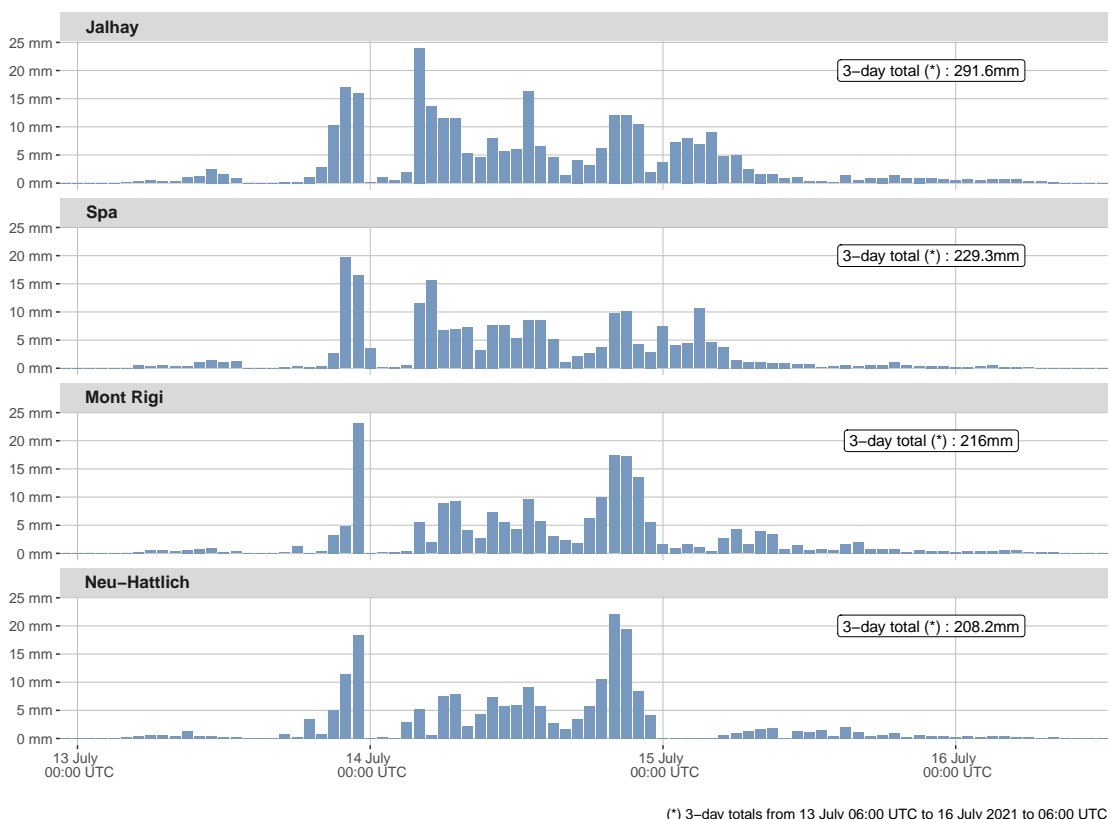

**Figure 7.** Time series of hourly precipitation totals for the 4 weighing rain gauges with a 3-day accumulation exceeding 200mm. These 4 rain gauges are located in the Province of Liège.

the results of Van de Vyver (2012) and Van de Vyver (2013). Figure 9 summarizes the frequency distribution of these 1-km sampled return periods for the 8 considered accumulation durations. The spatial distribution of the estimated return periods for each accumulation duration can be found in Figure E1. These results indicate that the precipitation accumulations for durations up to 3 hours did not reach exceptionally high values anywhere in Belgium (i.e., return period below 30 years). For longer

accumulation durations (from 6 hours to 3 days), a clear trend towards extreme values can be noticed with, in parallel, an increase in the size of the severely affected areas.

     Figure 9 illustrates the highly unusual nature of this event. In Belgium, extreme summer rainfall is generally caused by local convective storms and, consequently, extreme return periods are obtained for short durations and concern relatively small areas. In surrounding regions with similar climatic conditions, this type of event is also very rare. A relatively similar event

with mixed convective and stratiform rainfall occurred in the Netherlands and the bordering part of Germany in August 2021 (Brauer et al., 2011). Over an area of 740 km$^2$ more than 120 mm of rainfall were recorded in 24 h. Maximum amounts for the whole episode reached 160 mm, which is still lower than the extreme values obtained in the present case.

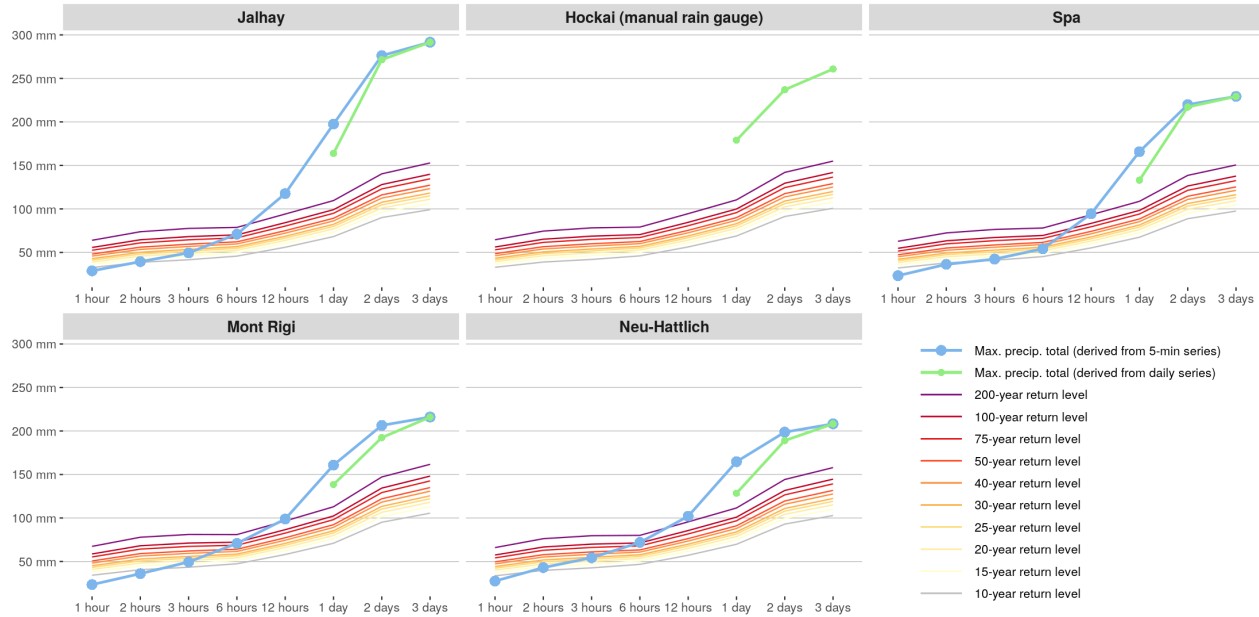

**Figure 8.** Maximum precipitation accumulation between 13th July 06:00 UTC to 16th July 2021 06:00 UTC for durations from 1 hour to 3 days for the 5 rain gauges with a 3-day accumulation exceeding 200mm compared against the 10-year to 200-year return levels. These maximum values are derived from 5-min time series (weighing rain gauges) as well as from daily time series (weighing and manual rain gauges). These 5 rain gauges are located in the Province of Liège. The return levels are derived from 10-min historical series for durations till 12-hours and from daily historical series for durations of 1 day or more (Van de Vyver, 2012, 2013).

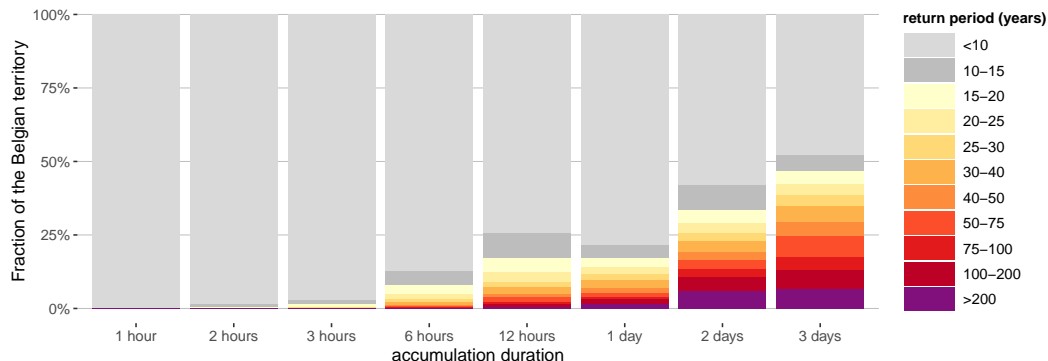

**Figure 9.** Frequency distribution on the Belgian territory of the return period associated to the maximum precipitation accumulation between 13th July 06:00 UTC to 16th July 2021 06:00 UTC for durations from 1 hour to 3 days. The maximum accumulations are derived from hourly (resp. daily) radar product for durations till 12 hours (resp. of 1, 2 or 3 days). The area of Belgium is 30688 km$^2$.

Long duration rainfall events, affecting large areas and producing flash floods in several catchments are relatively common in the Mediterranean and Alpine-Mediterranean regions as shown in several inventories of European flood events (Gaume et al., 2009; Marchi et al., 2010; Amponsah et al., 2018). However, these events occur mostly in Autumn under specific climatic forcing and triggering mechanisms that are not present in our region of interest.

The apparent discontinuity in Figure 9 from 12-hour to 1-day accumulations is related to the temporal granularity of the considered data: maximum accumulations are derived from hourly radar product for duration till 12-hours and from fixed daily accumulations (from 08:00am to 08:00am local time) for durations of 1, 2 or 3 days in order to be consistent with the extreme value statistics (Van de Vyver, 2012). The Hershfield factor (van Montfort, 1990) aimed to adjust the impact of the time series granularity when deriving extreme value statistics is not used to stay as close as possible to the actual statistics.

### 3.3 Spatio-temporal analysis of the heavy rainfall event

The rainfall distribution in time illustrated in Figure 7 is limited to the rain gauges with the largest 3-day accumulation, which are all located in the same area (i.e., East of the Province of Liège). Rainfall is however distributed in time differently in other areas of Belgium. The sequence of 5-min radar product (Goudenhoofdt et al., 2023) allows to grasp the spatial-temporal dynamic of the 3-day event. In complement, maps of 3-hourly precipitation totals derived from the 5-min radar product are provided in Figures E2 and E3. It remains however difficult to get an overall insight on the spatial and temporal evolutions of the rainfall patterns during the event from these 5-min or 3-hourly sequences.

Dimensionality reduction methods (Carreira-Perpiñán, 1997; Fodor, 2003; Cunningham, 2008) can be helpful in this context. The goal of dimensionality reduction is to provide a low-dimensional approximation of large data while minimizing the loss of relevant information contained in the data. These approaches aim thus at summarizing the essence of the data in a few representative variables. They provide an approximate but easier to interpret representation of the data. One of the most popular techniques for dimensionality reduction is principal component analysis (Jolliffe and Cadima, 2016; Wilks, 2011). In this analysis, because of the non-negative character of precipitation data, we will compute a non-negative matrix factorization (NMF) of the 5-min radar product. This NMF analysis will allow us to highlight that rainfall was distributed in time differently over the country. It will provide a synthetic view of the event from which qualitative descriptions can be derived. Background information regarding the NMF method are given in the Appendix D.

In order to analyze the 5-min rainfall by NMF, the data need first to be structured in a non-negative $m \times n$ matrix $V$ where each element $v_{ij}$ contains the radar value of the pixel $i$ for the timestamp $j$. This analysis is performed for the 1-km radar pixels located in Belgium (i.e., $m = 30669$) and the period July 13 00:00 UTC to July 16 12:00 UTC (i.e., $n = 1009$). Although NMF approximations of various ranks have been derived, the discussion will be focused on the rank-3 NMF that provides a clear decomposition of the data into 3 distinct spatio-temporal patterns illustrated in Figure 10. Each component $i$ is composed of a vector $\mathbf{w}_i$ with values for the $m$ pixels (i.e., spatial pattern of the component $i$) and a vector $\mathbf{h}_i$ with values for the $n$ timestamps (i.e., temporal pattern of the component $i$).

These results show that NMF provides a decomposition of the rainfall data into 3 factors with distinct spatio-temporal characteristics. The first factor corresponds to stratiform precipitation in the South East part of Belgium that started already in

the morning of July 13 and lasted rather continuously till the end of July 14. The second factor concerns the central part of Belgium with precipitation in the evening of July 13, followed by a rather dry situation on July 14 and again precipitation the whole July 15. Finally, the third factor extracts the heaviest precipitation pattern located in the Province of Liège. This pattern is characterised by successive periods with very intense convective precipitation that started in the evening of July 13 and lasted till the morning of July 15. The NMF analysis clearly shows that rainfall was distributed in time differently in various areas of the country. In view of temporal distributions that are overlapping for the three factors, it is hard to figure out what is the dynamical source of these patterns. This constitutes an interesting future research, going beyond the scope of the current data description.

The information given in Figure 11 indicates that the rank-3 NMF approximation provides a meaningful summary of the rainfall data. First, Figure 11A quantifies how the 3-day accumulation is approximated by the rank-3 NMF. The approximation error is rather low in most places: between -8% and 16% where the 3-day accumulation exceeds 100mm. There is however a region in central Belgium with 3-day accumulation (around 80mm in the radar product) that is largely underestimated by almost 50mm in the rank-3 NMF. This is a local pattern that is not captured by the rank-3 NMF but where the 3-day rainfall amounts remain rather moderate. In Figure 11B, the correlation between the radar product and the 3 temporal components $\mathbf{h}_i$ is assessed for each pixel. This analysis confirms that each of these temporal components are mostly correlated with the radar product in distinct geographical areas, which corresponds the spatial components $\mathbf{w}_i$. To conclude, the information provided by the 3 spatial components $\mathbf{w}_i$ and 3 temporal components $\mathbf{h}_i$ (i.e., the 3 maps and 3 time series of Figure 10) summarizes well in an easily interpretable way the rainfall data for areas with a 3-day accumulation above 100mm.

## 3.4 Analysis of areal averages

Estimation of areal rainfall averages for catchments may be useful of the hydrological analysis of the event. Such areal averages can easily be derived from the radar product by considering the mean value of all pixels included in the domain of interest. Figure 12 illustrates hourly timeseries of areal rainfall averages for 6 catchments of different size (defined in Figure 1). Areal averages derived from the radar product are compared against an alternative estimation solely based on rain gauge data, i.e., an IDW interpolation of hourly rain gauge data at 1km resolution followed by a spatial aggregation of the interpolation points included in the respective domains. The comparison between both estimations indicates that their difference increases for smaller catchments. Figure 12 indicates also a very large variability in time of hourly rainfall for the Vesdre, Hoëgne and Wayai catchments. This variability in time is significantly smoothed for averages over the large-sized Meuse catchment. By spatially and temporally integrating the radar product, the total precipitation amount that fell over the Belgian part of the Meuse catchment during the 3 days is estimated to 1.527 km$^3$ (110 mm). Similarly, the total precipitation over the Ourthe and Vesdre catchments is estimated to 0.240 km$^3$ (130 mm) and 0.125 km$^3$ (180 mm), respectively.

The maximum accumulation for durations from 1 hour to 72 hours derived from these hourly areal time series is provided in Figure 13 for the 6 considered catchments. These values increase significantly between 6 hours and 48 hours of accumulation duration. The 48-hour totals are estimated to 100mm on average over the large-sized catchment of the Meuse and reach almost

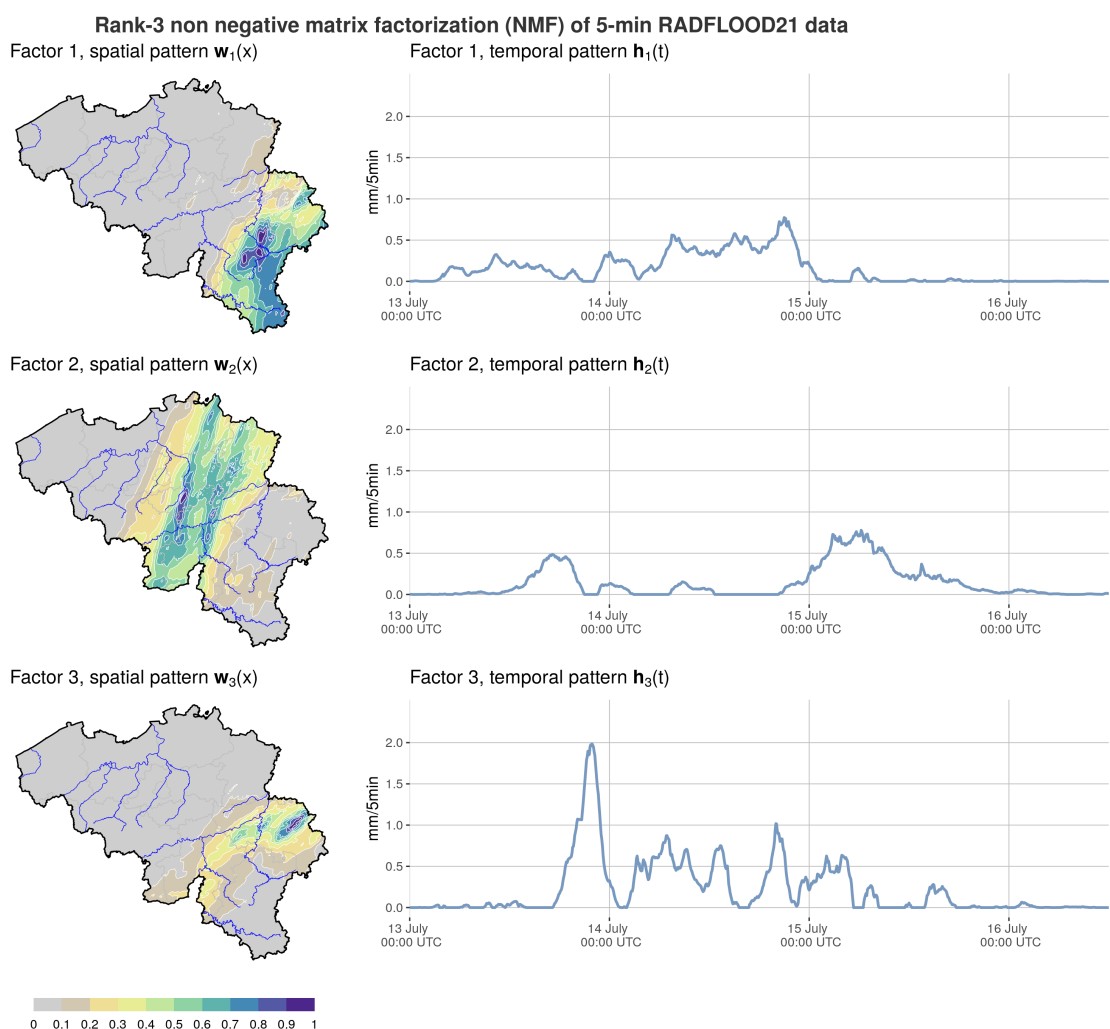

**Figure 10.** Spatial and temporal patterns of the 3 components resulting from a rank-3 NMF of the 5-min rainfall data.

200mm on average for the 2 smallest catchments of the Hoëgne and Wayai. Cumulative precipitation sums derived from the hourly timeseries in Figure 12 are illustrated in Figure E4 for the interested readers.

Finally, in order to quantify how the difference between the IDW of rain gauge data and the radar product evolves when they are aggregated in space and time, we considered 67 hydrological catchments in Belgium of various sizes with an average 3-day accumulation exceeding 20mm. Hourly areal averages were derived for each catchment and then accumulated for various durations till 72h. For each catchment and accumulation duration, the difference between all pairs of IDW and RADFLOOD21 estimates is characterized by the mean absolute error (MAE) normalized by the mean of the RADFLOOD21 values during the 3-day period. Average MAE values for catchments of similar sizes are finally computed and illustrated in Figure 14. These

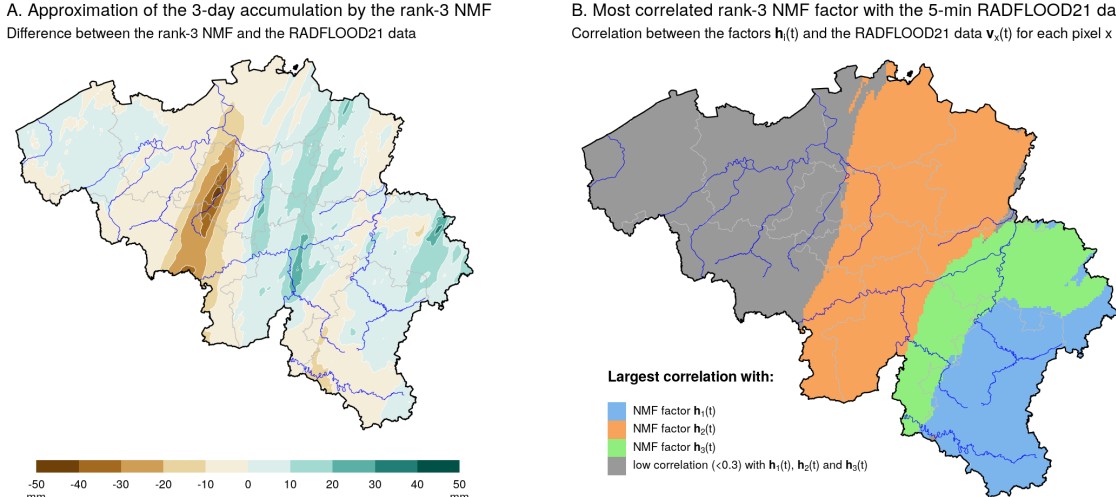

A. Approximation of the 3-day accumulation by the rank-3 NMF
Difference between the rank-3 NMF and the RADFLOOD21 data

B. Most correlated rank-3 NMF factor with the 5-min RADFLOOD21 data
Correlation between the factors $\mathbf{h}_i(t)$ and the RADFLOOD21 data $\mathbf{v}_x(t)$ for each pixel x

Largest correlation with:
- NMF factor $\mathbf{h}_1(t)$
- NMF factor $\mathbf{h}_2(t)$
- NMF factor $\mathbf{h}_3(t)$
- low correlation (<0.3) with $\mathbf{h}_1(t)$, $\mathbf{h}_2(t)$ and $\mathbf{h}_3(t)$

**Figure 11.** A. Difference on the 3-day accumulation between the rank-3 NMF and the RADFLOOD21 data.

B. Result of the correlation analysis between the rainfall data and the 3 temporal components $\mathbf{h}_i$ for each pixel: the map illustrates which component $\mathbf{h}_i$ is the most correlated with each pixel data, provided that the correlation is larger than 0.3.

results clearly highlight a decrease of the discrepancies between both datasets when the data is aggregated in space over larger domains and accumulated over longer periods.

## 4 Conclusions

The rainfall event of July 13 to 16, 2021 over Central Europe affected considerably the eastern part of Belgium and caused disastrous floods in several river catchments. In order to understand the course of the events and the various factors relating rainfall to flood response, it is essential to obtain the best picture of the spatio-temporal distribution of rainfall. In the present work, we present and we analyse quality controlled rain gauge observations and radar-based rainfall products. The radar products include a merging with automatic gauge measurements. These data are freely available and provided as a supplement on Zenodo (Journée et al., 2023; Goudenhoofdt et al., 2023).

The most extreme rainfall were observed in areas with high orography and estimating rainfall from radar observations is particularly challenging for this event. Radar ground echoes are more frequent and intense in areas with high orography which affects the quality of the raw radar data. Orographic enhancement is suspected to have played an important role, which makes it important to use radar observations as close to the ground as possible. A proper treatment of ground echoes was therefore essential for this case.

In some areas, spatial structures at fine scale present in the radar-based products differ substantially from those obtained by classical interpolation of rain gauges. Verification shows that errors made by the radar-based product are considerably

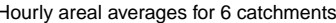
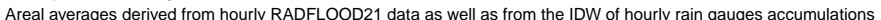
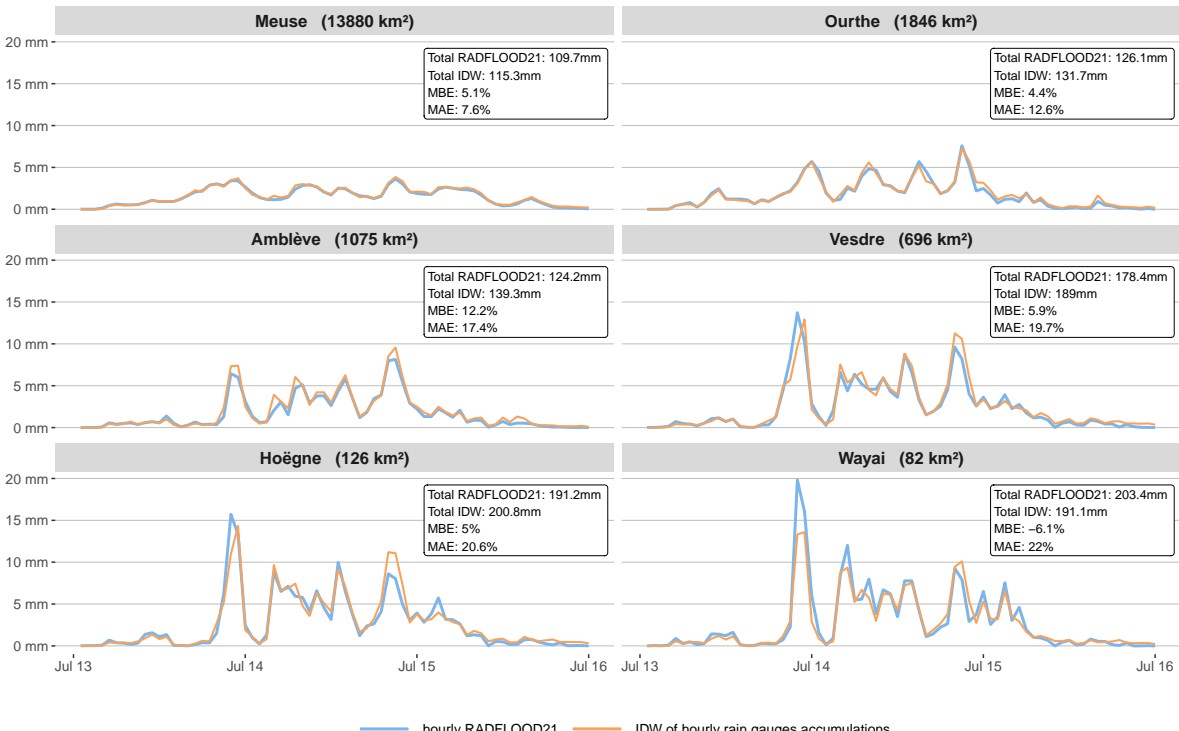

**Figure 12.** Hourly timeseries of areal rainfall averages for 6 catchments of different size which are mapped in Figure 1. The hourly areal averages are derived from the hourly radar product as well as from hourly rain gauge data (i.e., IDW interpolation of rain gauge data at 1km resolution followed by a spatial aggregation of the interpolation points included in the respective domains). The difference between both estimations is summarized by the mean bias error (MBE) and the mean absolute error (MAE). The MBE is computed as the average difference between the hourly IDW and RADFLOOD21 estimates (i.e., IDW-RADFLOOD21). Both MBE and MAE are normalized by the average RADFLOOD21 value over the displayed period.

smaller than those obtained from rain gauge interpolation. For small catchments, incorporating radar observations is required for capturing the temporal evolution of rainfall.

A generic statistical analysis of these data is also provided allowing for characterizing the return period of such an event and the spatial structure of the precipitation field. This analysis shows that the event of mid-July 2021 is a record-breaking event over most of the eastern part of Belgium with return periods largely exceeding 50 years. The 200-year return level is exceeded over 2000 km$^2$. The most extreme rainfall over 2 days is almost twice the 200-year return level. The analysis of the spatial structure of the rainfall field suggests that there are three dominant spatial structures over Belgium. As these are not well separated in time, additional analyses are needed in order to relate these features to physical processes. Regarding total

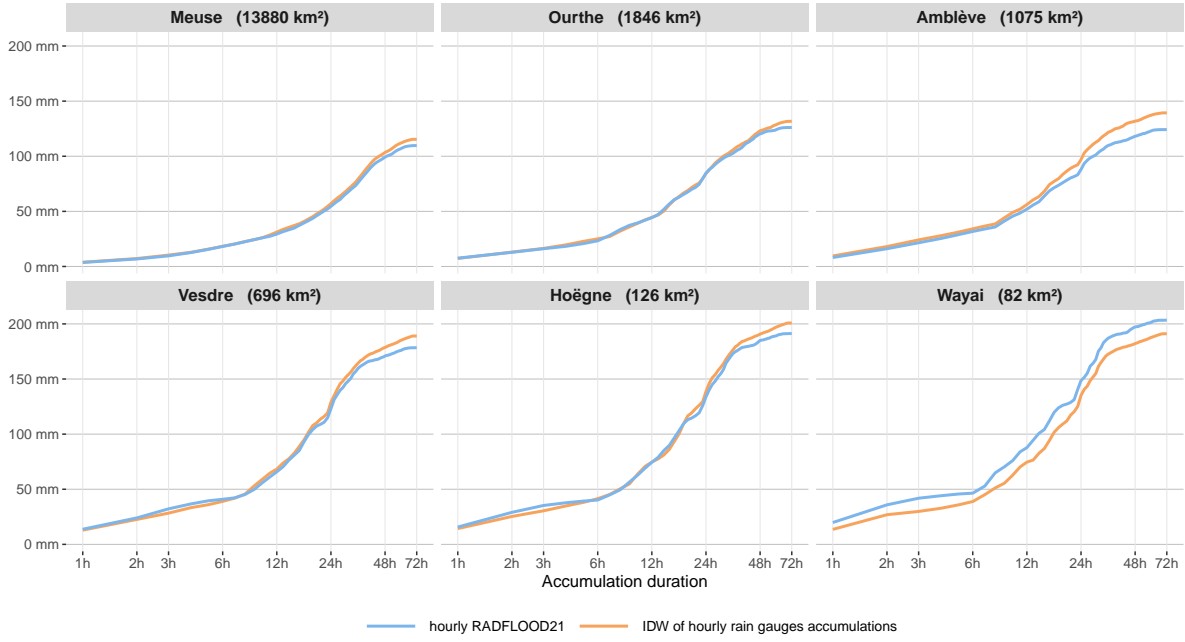

Maximum precipitation accumulation between 13th July and 16th July 2021 for durations from 1 hour to 72 hours (3 days) derived from hourly areal averages for 6 catchments

**Figure 13.** Maximum precipitation accumulation for durations from 1 hour to 72 hours by steps of 1 hour derived from hourly timeseries of areal rainfall averages for 6 catchments of different size which are mapped in Figure 1. The hourly areal averages are derived from the hourly radar product as well as from hourly rain gauge data (i.e., IDW interpolation of rain gauge data at 1km resolution followed by a spatial aggregation of the interpolation points included in the respective domains).

precipitation amounts integrated over a region during the event, quantities of 0.484 km$^3$ (125 mm) and 1.527 km$^3$ (110 mm) are estimated for the Province of Liège and for the Belgian part of the Meuse catchment, respectively.

It is also interesting to note that the return periods are relatively moderate for shorter durations than 6 hours. Rainfall
temporal evolution can not be considered as typical of a flash flood event. In contrast, the response of the system appears, according to several testimonies, as very close to flash flood with surprise effects at various levels (Van Camp et al., 2022; Dewals et al., 2021). This points out the high non-linearity of the system and the fact that various other catchment-specific parameters like topography, land use or soil properties played an important role. Additional processes like the transport of debris by fast-flowing water need to be considered to understand the relation between precipitation and flood response. The
presence of dams in some valleys affected by floods should also be taken into account.

The data sets presented here have been produced and quality-controlled with the utmost care. Nevertheless, some additional work can be performed to further refine rainfall estimates. The radar-based rainfall product used in the present study is based on single-polarization radar information only. The exploitation of polarimetric information available from the dual-polarization radars covering the area of interest could bring some additional benefits. Extending the area of interest, incorporating additional

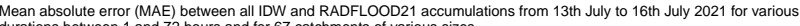

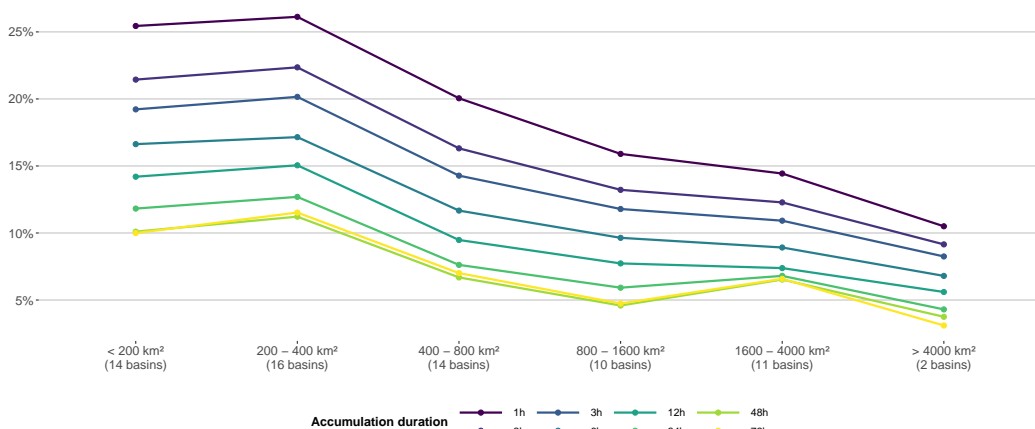

**Figure 14.** Mean absolute error (MAE) between all IDW and RADFLOOD21 accumulations from 13th July to 16th July 2021 for various durations between 1 and 72 hours and for 67 catchments of various size in Belgium (from 61 km$^2$ to 14800 km$^2$). The 3-day RADFLOOD21 accumulation exceeds 20mm on areal average for all these catchments. The MAE is normalized per catchment and per accumulation duration by the average RADFLOOD21 value over the 3-day period. Average MAE values for catchments of similar sizes are finally displayed in the figure.

information from radars and rain gauge networks available in neighbouring countries, could be realized within an international effort for producing the best picture of this event over the full area affected by floods.

The usage of high-quality rainfall data of extreme events is of great interest interest in many fields. Besides the question of the impact of climate change on the occurrence of such events, many questions arise related to water resource and flood risk management or land use planning. Understanding the full causality chain from rainfall to floods and further to the resulting im-

pact in terms of casualties and economic losses is a challenging multidisciplinary enterprise. The present work is a contribution to this global effort.

*Data availability.* The discussed rain gauge data and radar product RADFLOOD21 can be accessed on Zenodo (Journée et al., 2023; Goudenhoofdt et al., 2023).

## Appendix A:  Filtering of static non-meteorological radar echoes

### A1  Doppler filtering

The radar signal processor includes a Doppler filter for removing clutter. The Wideumont radar filter is a time domain Infinite Impulse Response filter. The more recent Jabbeke and Helchteren radars include a frequency-domain filtering which computes

| reflectivity (dBZ) | 0 | 5 | 10 | 15 | 20 | 25 | 30 | 35 | 40 | 45 | 50 | 55 | 60 | 65 |
|---|---|---|---|---|---|---|---|---|---|---|---|---|---|---|
| frequency | 0.2 | 0.2 | 0.2 | 0.2 | 0.18 | 0.16 | 0.14 | 0.1 | 0.04 | 0.02 | 0.01 | 0.005 | 0.005 | 0.005 |

**Table A1.** For a given reflectivity level, the minimum frequency which will never be reached by precipitation only under the current Belgian climatic conditions.

the power spectrum using a Discrete Fourier Transform (DFT). The time domain filter only suppresses the signal in the filtered region while the DFT filter is also designed to restore the weather signal in the filtered region. In addition, the processing chain includes threshold checks to remove data of low quality. The clutter correction ratio (CCOR) is defined as the logarithmic ratio between the unfiltered signal power and the clutter filtered signal power. The CCOR for each range gate is compared with a user set threshold and data with a CCOR above this threshold are set invalid.

## A2 Computation of the static clutter map

A measurement location is considered systematically contaminated by non-meteorological echoes (clutter) if:

- its height is below 1200 m above the radar level

- its distance from the radar is below 100 km

- its frequency of exceeding 7dBZ is above 20% for a given period

For the sake of consistency, the static clutter map is smoothed to remove small holes with valid data.

## A3 Computation of static clutter level statistics

The frequency distribution of the reflectivity is computed by bins of 5dB.

In most cases, the fixed target causes a signal with a relatively constant reflectivity. This corresponds to the bin with the highest frequency. It is considered as the mean clutter level.

Some targets, like wind farms, can cause a fluctuating signal, and the mean statistics is useless. For a given reflectivity bin, one can determine if its frequency is unrealistic in the current Belgian climatic conditions. By taking the upper bound of the highest unrealistic bin, one obtains the maximum detectable clutter level.

The unrealistic frequencies have been optimized by trial and error using several years of data. The results can be found in Table A1.

## A4 Identification of static clutter

A reflectivity measurement is considered as static clutter if:

- it belongs to the static clutter map computed in the last two months

- its value is not 5 dB above the maximum clutter level computed in the last two months

Therefore an actual rainfall value is kept if it is sufficiently large.

## A5    Handling of false zeros

The CCOR threshold is set to 30 dB for the Jabbeke and Helchteren radars and 15 dB for the Wideumont radar. When the
Doppler filter removes more than this threshold the reflectivity is improperly set to "undetect" by the radar processing (which
means zero precipitation). Other kind of processing from all radars are also introducing false zeros. This is problematic since
the static clutter statistics cannot be computed reliably anymore. The only guaranteed solution is to use a static clutter map and
maximum clutter level based on unfiltered reflectivity.

Using the unfiltered statistics results in a significant reduction of the filter performance. To keep some efficiency, a value
which is valid for the filtered statistics and which exceeds the unfiltered mean clutter level minus the CCOR threshold is not
rejected.

Some residual clutter might remain at this stage but the clutter processing further includes dynamical filters based on satellite
cloudiness, vertical profile of reflectivity and spatial texture (see Section 2.2.2).

## A6    Results

Figures A1 and A2 show an example of the static clutter map and statistics for the radar of Helchteren, valid during the flood
event, in the Vesdre catchment. It can be seen that a significant part of the area is affected by strong clutter. The Doppler
filtering is very efficient and some high precipitation values can be retrieved.

## Appendix B:  Filtering of dynamic non-meteorological radar echoes using vertical radar information

A measurement at a given elevation is considered as clutter (non-meteorological) if the gradient between its value and the
corresponding (horizontally) interpolated value on a higher (lower) elevation exceeds in magnitude -20 dBZ km1 (+10 dBZ
km1). Because of variations from signal fluctuations, a minimum absolute difference of 5 dBZ between two corresponding
values at different elevations is required for clutter identification.

## Appendix C:  Tuning of the KED method

The KED method has been tuned on Belgian rainfall. Here are the main parameters:

- Interpolation of the 4 closest radar values at gauge location

- Spherical covariance model for the residual error with a range of 20 km

- Correlation limited to 70 % in a 1 km neighborhood due to sampling and measurements errors

- Square root transformation to ensure Gaussianity

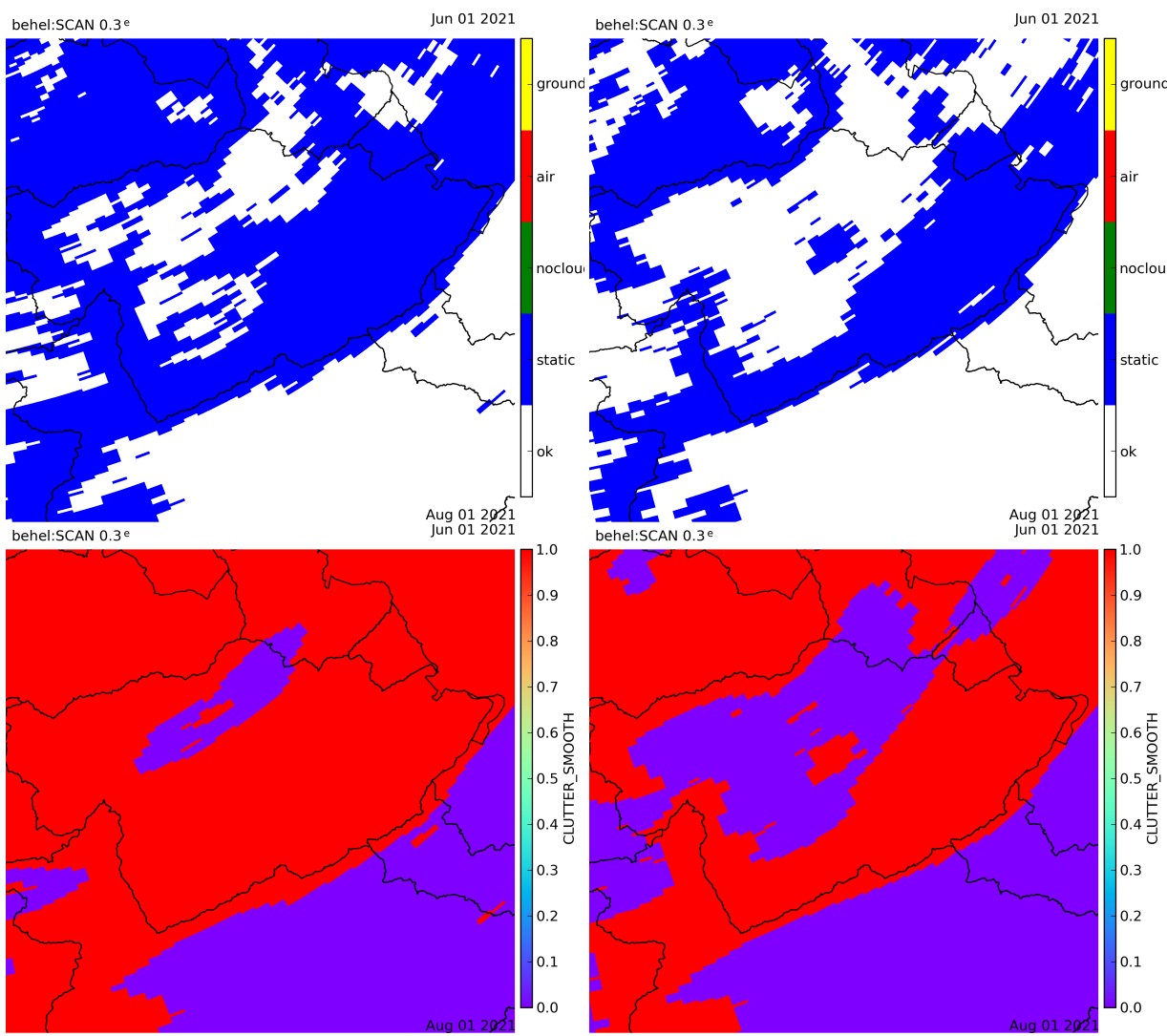

**Figure A1.** Static clutter map for the unfiltered (left) and filtered (right) reflectivity, before (top) and after smoothing (bottom). The map is centered on the Vesdre catchment.

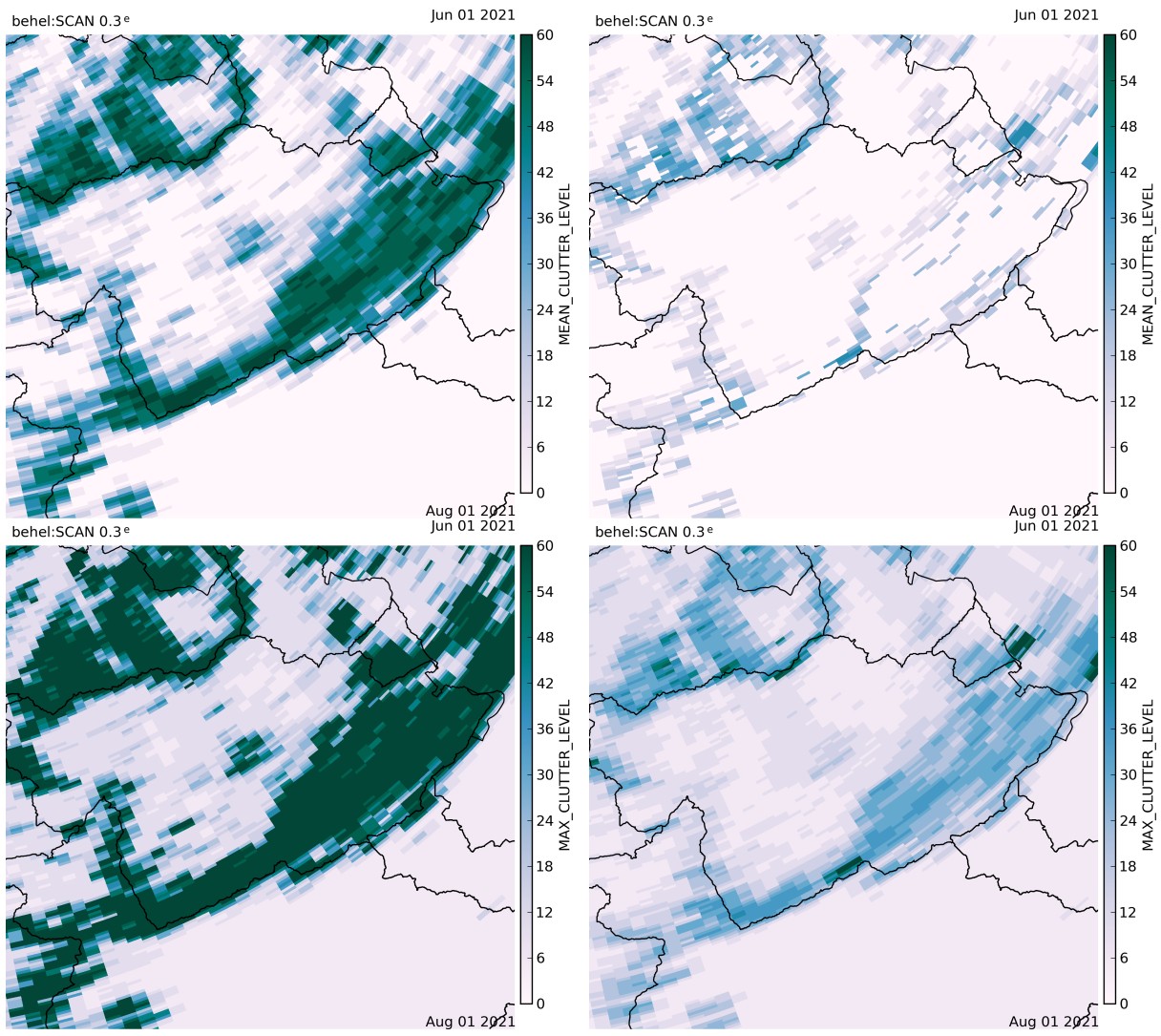

**Figure A2.** Mean (top) and maximum (bottom) clutter level for the unfiltered (left) and filtered (right) reflectivity. The map is centered on the Vesdre catchment.

- Use of the 21 nearest gauges for a given location

- Not valid locally if less than 4 gauge values above 0.1mm

- Not valid locally if the correlation ($R^2$) between radar and gauge is below 0.5.

- Invalid data replaced by a single gauge bias correction

- Smooth transition to a single gauge bias correction outside the convex hull

**Appendix D:  Non-negative matrix factorization (NMF)**

Given a data matrix $V$ with $m$ rows and $n$ columns (i.e., a matrix of dimension $m \times n$) and with non-negative elements $v_{ij} \geq 0$ (i.e., $V$ is a non-negative matrix), the non-negative matrix factorization (NMF) approximates $V$ as the product of two non-negative matrices $W$ and $H$ of dimensions $m \times k$ and $k \times n$ respectively,

$$V^{(m \times n)} \approx W^{(m \times k)} \, H^{(k \times n)}. \tag{D1}$$

The approximation (D1) of $V$ can equivalently be written with vectors instead of matrices by

$$V^{(m \times n)} \approx \sum_{i=1}^{k} \mathbf{w}_i^{(m \times 1)} \, \mathbf{h}_i^{(1 \times n)}, \tag{D2}$$

where $\mathbf{w}_i$ (resp. $\mathbf{h}_i$) of dimension $m \times 1$ (resp. $1 \times n$) denotes the $i$th column of $W$ (resp. the $i$th row of $H$). Equation (D2) formulates the approximation of $V$ as the sum of $k$ matrices of rank 1 (i.e., the matrix product of 2 vectors), which are called *components* or *features*. In practice, $k$ is set much smaller than the dimensions $m$ and $n$ in order to decompose $V$ into a limited number of components expected to represent meaningful characteristics of the original data. NMF is generally solved as an

optimization problem, where an objective function characterizing the difference between $V$ and matrix product $WH$ has to be minimized. Various algorithms for NMF have been proposed in the literature. In this analysis, we used the algorithm rNMF (Sun et al., 2013) available as an R package. We refer to Gillis (2020) for more detailed information about NMF.

## Appendix E: Additional figures

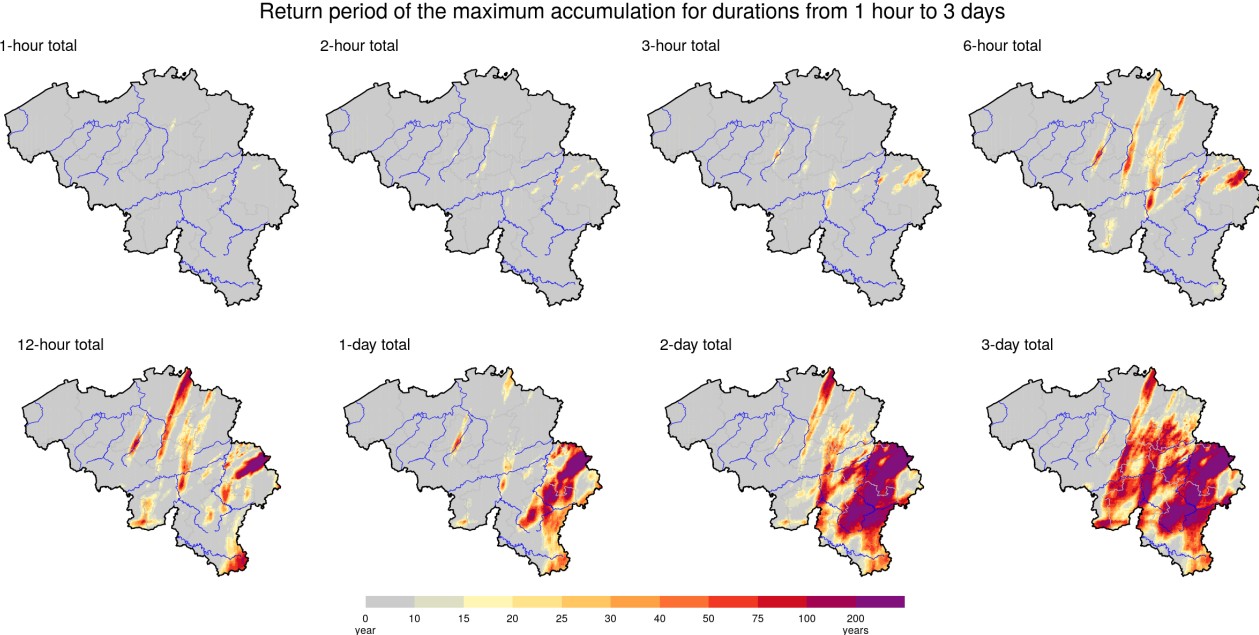

**Figure E1.** Return period estimated for the maximum precipitation accumulation between 13th July 06:00 UTC to 16th July 2021 06:00 UTC for durations from 1 hour to 3 days. The maximum accumulations are derived from hourly (resp. daily) radar product data for durations till 12 hours (resp. of 1, 2 or 3 days).

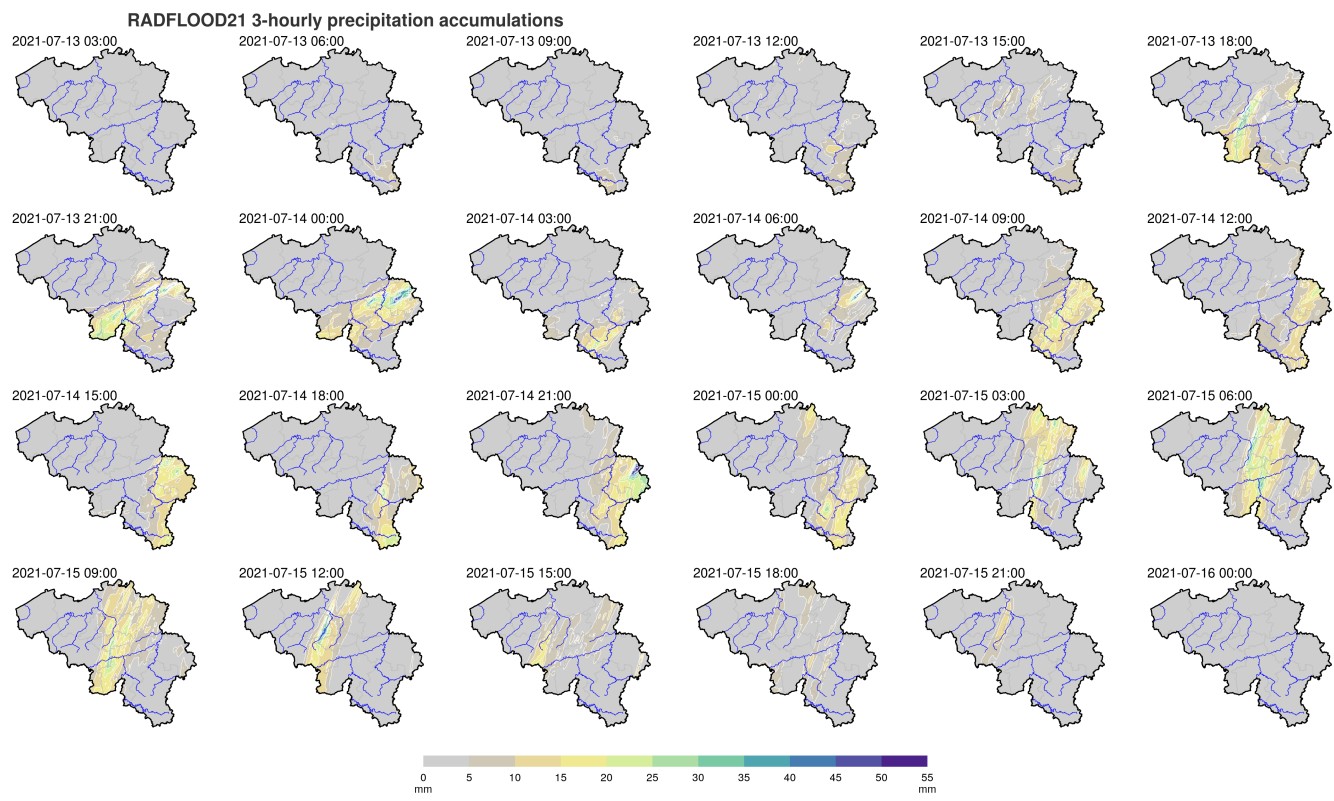

**Figure E2.** 3-hourly precipitation accumulations in Belgium derived from 5-min radar product data. The timestamp on top of each map indicates the end of the 3-hour period in UTC.

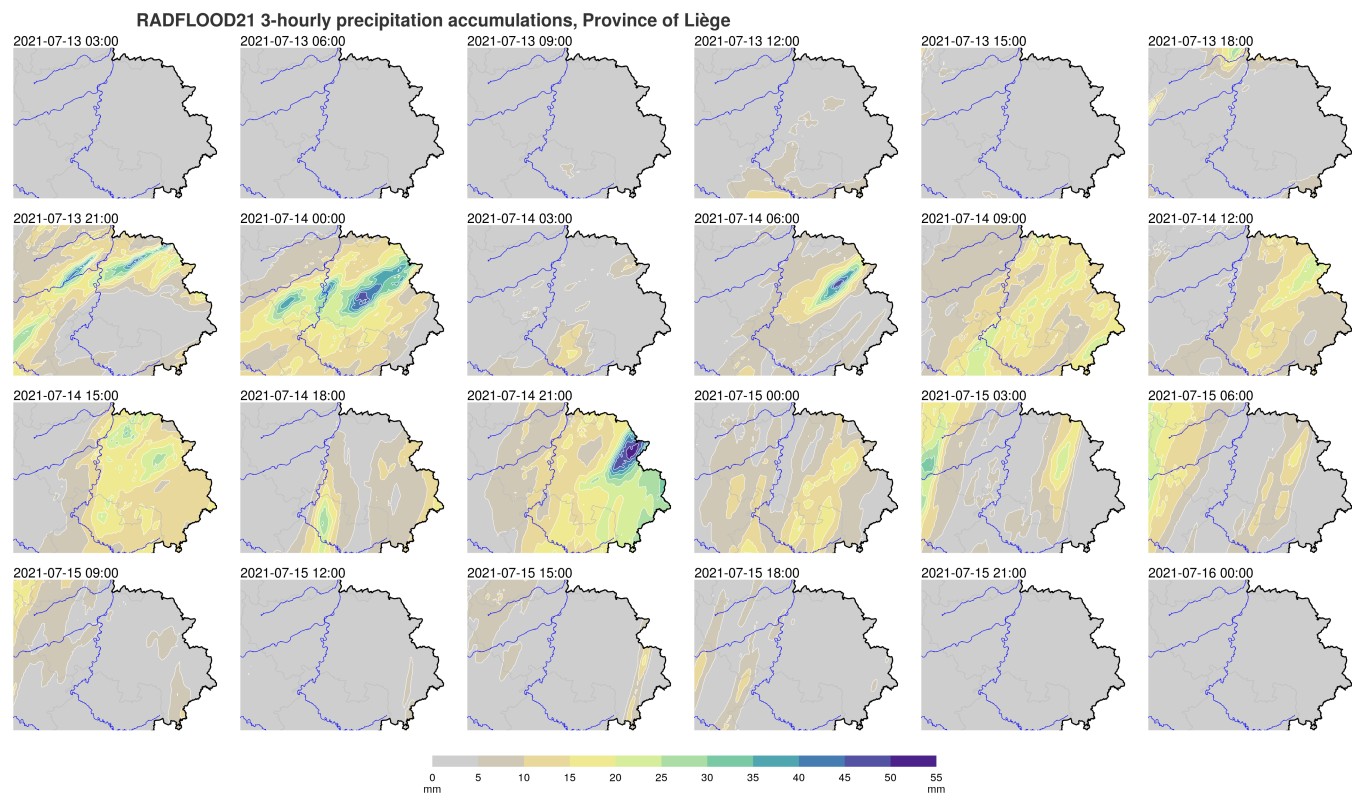

**Figure E3.** 3-hourly precipitation accumulations in the Province of Liège derived from 5-min radar product data. The timestamp on top of each map indicates the end of the 3-hour period in UTC.

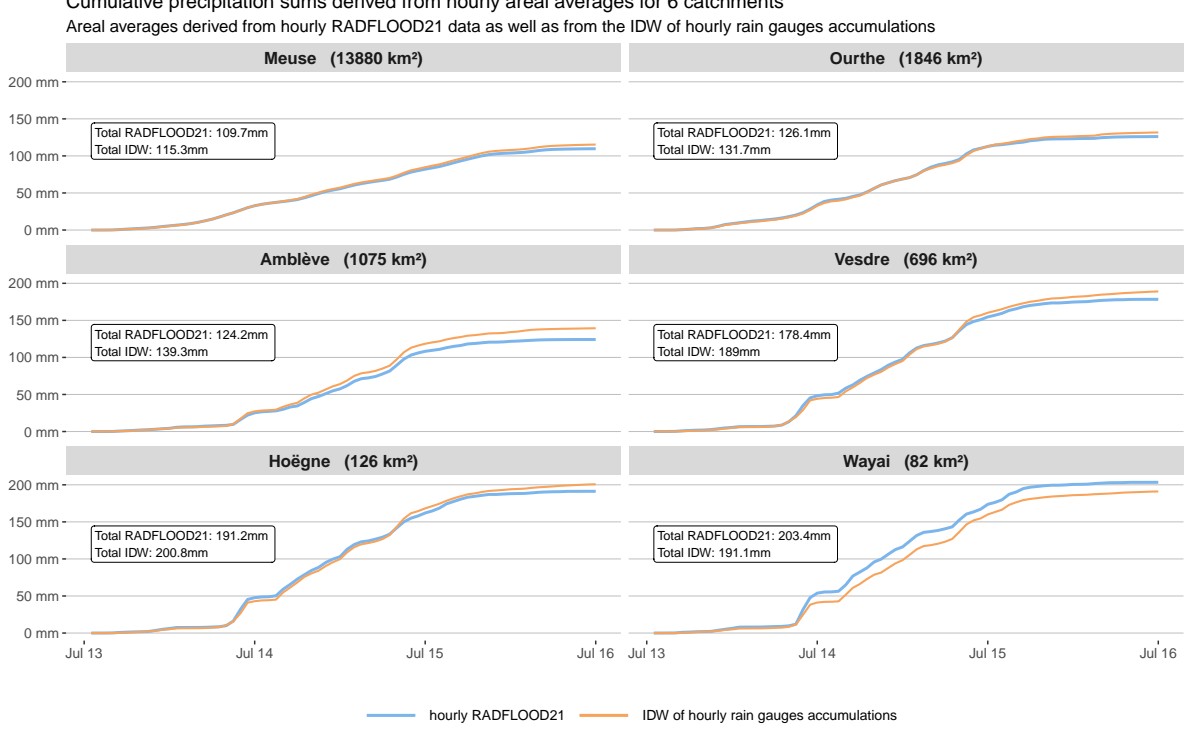

**Figure E4.** Cumulative precipitation sums derived from the hourly timeseries of areal rainfall averages illustrated for 6 selected catchments in Figure 12.

*Author contributions.* MJ extracted and verified the rain gauge data; EG developed the radar product RADFLOOD21; MJ analyzed the data;
450  MJ, EG, SV and LD wrote the manuscript.

*Competing interests.* The authors declare that they have no conflict of interest.

*Acknowledgements.* The authors thank Ruben Imhoff and Wolfgang Wagner for their careful review of the manuscript. Their construc-
tive comments and suggestions were greatly appreciated. This study makes use of recordings from rain gauges operated by the following
Belgian regional hydrological services: Service Public de Wallonie - Mobilité et Infrastructures (https://hydrometrie.wallonie.be), Vlaamse
Milieumaatschappij (https://www.waterinfo.be/) and Hydrologisch Informatiecentrum (https://www.waterinfo.be/). Radar observations from
Vlaamse Milieumaatschappij, German Weather Service and Météo-France were used as well. All teams responsible for the maintenance and
operation of rain gauge and radar networks in Belgium and the neighbouring countries are warmly thanked.

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
