# Peer review of "Quantitative rainfall analysis of the 2021 mid-July flood event in Belgium"

_EGUsphere, 2023_

## Referee Comment (RC1)

Review of: **Quantitative rainfall analysis of the 2021 mid-July flood event in Belgium** by Michel Journée et al.

Ruben Imhoff

Ruben.Imhoff@deltares.nl

April 29, 2023

**Summary**

The authors present an extensive analysis of the rainfall that has fallen over Belgium, and in particular the heavily hit Ardennes region, during the July 2021 flooding event. The authors introduce two rainfall products: one based on rain gauges, while the other product is the most recent radar QPE product of the Belgian RMI that has been improved with, among others, the information of this event. Both products are described in this paper, are analyzed and made openly available. I think and do agree with the authors that this rainfall dataset, including the analyses, is relevant for the extreme flooding event and potential follow-up studies that can make good use of such a validated rainfall dataset. However, I think that the manuscript in its current shape is not clear enough about what really is new in the processing and derivation of the rainfall datasets compared to what is available anyway. Hence, I would like to see somewhat more emphasis on the (new) methods used for these datasets, particularly for the radar QPE. In general, the methods can be described in more detail throughout the paper.

In addition, the structure of the paper was not always clear to me, as many methods are described in the results section. The paper would become clearer when all methods are placed in the methods section and when both the rainfall estimation and analysis methods are described in more detail. Finally, I think that the rainfall analysis for the (hydrological) application area(s) can be further emphasized and, where possible, extended. As this will be the application scale and field of the outcomes of this paper, I think this would increase the impact of the work. Summarizing, I think the paper needs some restructuring and more elaborate descriptions of the methods. The analyses themselves seem appropriate and supportive of the type of work and drawn conclusions.

In the sections below, I describe my suggestions in more detail in the general comments and line-by-line in the specific comments. Some minor suggestions for technical corrections are placed at the end under 'technical corrections'. I hope it helps in further improving this manuscript, which I hope to see in a published form in due course.

**General comments**

*Rainfall product description*

Besides the analyses of the quality of the rainfall products, I think a major component of this paper is the upgrade of the rainfall products and the fact that they are openly available for this event now. The descriptions of the rainfall product creation, especially for the radar QPE, is sometimes a little too brief and gives the impression that not much changed compared to the data that was already

present. I think that this does not give the work the rightful weight (and credit) on the improvements that have been made to come up with a decent radar QPE product, which is very challenging for a combination of an extreme event and (sometimes) orographically enhanced rainfall. Hence, I am fully supporting the enhancements that the authors have made to, in particular, the radar QPE product, but the description could be more extensive to make it clearer what processing steps have been taken and what has been changed compared to the operationally available product at that time. In the specific comments below, I have indicated where the authors can extend the descriptions.

*Description of the methods*

In line with my previous comment, I have noticed that the methodological description is sometimes too brief, which does not make it directly clear to the reader how certain analyses are performed and why certain choices were made. In the section with specific comments, I have mentioned where I think this is the case and what, in my opinion, can be added to the methods description to make it clearer.

In addition, the results section contains many methodological descriptions. It makes more sense to put the methodological descriptions that are present in the sub sections of the results in section two. This makes the paper better structured, which also helps in directly grasping the extend of the analyses that are present in this paper. When I started reading the paper, it seemed like it was only focusing on the creation of the rainfall products, but there turned out to be way more to it (so, a better structure helps there).

*Application of areal averages (section 3.4)*

This section does not get the weight it deserves, in my opinion. This is the application scale and, I think, one of the reasons the authors have made this dataset available and have put this analysis on paper. I think it would be good to extend this analysis a bit and show more hydrologically relevant information. The focus on the different catchment sizes is, by the way, very relevant and interesting.

The extension of the analyses could consist of emphasizing the cumulative rainfall sums (partially already present, but see my comments about figure 13 later on), as this directly shows both the difference in rainfall volume of the time period and when these differences occurred, increased or decreased. In addition, it would be even more interesting to see the effect of these differences on the simulated discharge for these catchments. I am aware that this may be slightly outside the scope of the current study, but by, for instance, applying the operationally used hydrological model with these two datasets as input, the authors could already present a very simple estimate of the effects of the different rainfall products on the simulated discharge. Do not see this as a must, but it would increase the impact of the work.

**Specific comments**

Lines 27 – 29: What does this exactly mean for hydrology and/or the flooding that took place?

Lines 70 – 74: What was the density (per $km^2$) of the gauging network and how does this density differ per region. I.e., how is the density in a heavily hit catchment such as the Vesdre, for instance? It's sometimes hard to estimate that from the figure.

Lines 81 – 82: How did this gap filling exactly work? Were gaps replaced with the data of the closest station, was some kind of spatial interpolation method used or something else?

Lines 82 - 84: How were the daily values adjusted? This, as well as my previous comment, is quite relevant information for the dataset you are providing.

Line 104 – "different kind of technology": do you mean that the hardware of the radars is different?

Lines 104 – 105: This is indeed a major challenge. It would be interesting for the reader if the authors can briefly describe how RMI handles this to make their composite.

Lines 109 – 112: Could the authors describe in a little more detail what this basic radar estimation method is (just the Marshall-Palmer reflectivity to rain rate transformation, or more?) and how the comparison with the rain gauges took place. Regarding the latter, we often compare gauge location by corresponding grid cell, but it is also possible to take the rainfall advection into account (as the radar and rain gauges measure at different elevations), etc.

Lines 113 – 120: I think this is a major improvement of the radar rainfall estimation in hilly regions. Actually, this improvement makes the provided and described dataset unique and different from what was already available at the time. Although I am familiar with the method used, I do not think that the average reader is. Hence, can I ask the authors to put more emphasis on this procedure and describe it in more detail.

Lines 123 – 124: What was considered abnormal or unrealistic, i.e. was there a threshold or is this qualitative?

Lines 129 – 130: How is this identification of the precipitation type exactly done and what Z-R relationship is used? I know most of this is described in Goudenhoofdt and Delobbe (2016), but either refer explicitly to this paper or (/and) describe it here.

Lines 136 – 137 - "The rainfall accumulation over 5 minutes is obtained by computing the movement of precipitation using optical flow techniques": What kind of optical flow techniques were used?

Lines 139 – 143: If any methods from other papers were used in the KED method and setup, then some referencing is necessary here.

Lines 160 – 161: Although I fully agree, what can be said about the quality of the radar here? I.e., is the radar missing any precipitation in space, or overestimating, in this region? As radar estimates in hilly and mountainous areas are challenging, this is to be expected. It would be very interesting to have an idea of this too. This could also point out why and where the combination of both data sources is going to be crucial for a good analysis of this event. The answer to this comes already partially back in Figure 4, so good to give some more details here.

Lines 164 – 168: This actually belongs in the methods section, which is right now missing for the analysis part of this paper.

Lines 175 – 176: Could the authors briefly describe the used method (and reason to use that method) from Van de Vyver (2012, 2013) in the methods section and then refer to the paper?

Lines 206 – 208 – "For longer accumulation durations (from 6 hours to 3 days), a clear trend towards extreme values can be noticed with, in parallel, an increase in the size of the severely affected areas.": I think the authors can put some more emphasis on this finding. It would be interesting to relate this to literature findings (either here on in a discussion section) for similar extreme flooding events or similar rainfall patterns. I.e., is this what you expect or is this unique to this event?

Lines 221 – 243: This belongs in the methods section.

Section 3.3: I am not familiar with the NMF method, but can I ask the authors to explain a bit more elaborately what you are trying to reach with this method and why you focus in the results on a rank-3 NMF (and not more or less)? Why do we need the NMF method? Is it to be able to split the mesoscale rainfall field into more regional scale stratiform and convective rainfall, or is there another reason?  See also my comment below related to figure 10a.

Figure 3: Here I would show the catchments that you are focusing on instead of the province of Liège.

Figure 10a: The sub panel is showing the difference between the rank-3 NMF and RADFLOOD21 data, right? This is not yet clear from the caption. In addition, from the figure and the corresponding text it is not directly clear to me if a difference is good or bad and how big of a difference is considered acceptable. As mentioned earlier, I am not familiar with the NMF method, so that definitely plays a role too. Is a small difference and indication that with (just) three ranks, and corresponding regions, you capture most of the rainfall in space and time, or should I interpret it differently?

Figure 12: Useful figure, but I think this figure fits better in the study area description. Hence, put it together with one of the first figures. I think it would fit well in figure 3.

Figure 13: It looks like the individual points in the graphs have relatively coarse steps (1, 2, 3, 6, 12, etc. hours), even though the data is based on hourly accumulations. Hence, I think the authors could put more data points in this graph, which more clearly identifies when differences between the two datasets occur. In fact, why not just go to 5-min steps and derive the cumulative sums on that time step?

Figure A1: Just to double check, we are looking at the maximum x-hour window accumulation that occurred, right? So, for e.g. the grids cells in the Vesdre, we are looking at the statistics of the maximum x-hr accumulation that occurred during the 3-day period and not the average? Besides, do you look at a rolling window in which for every 5-min steps a new x-hour window is assessed or have the authors assessed it differently? Good to briefly mention this in the methods and caption.

**Technical corrections**

Line 1 – "impacted severely Belgium": impacted Belgium severely.

Line 20 – "total cost": total economic damage (would suit better, I think).

Line 43 – "analysis": analyses.

Line 44 - "event": events.

Line 86 – "details": detail.

Line 107: Although true, I think this needs some referencing.

Lines 130 – 131: Good to add a reference here.

Line 134 – "are here used"; are used here.

Line 157 – "is areas": in areas.

Line 216 – "in a same area": in the same area.

Line 225 – "technique": techniques.

Lines 275 – 278: From a hydrologist's perspective, it is more useful to provide these numbers in mm instead of in km$^3$. The same comment holds for the conclusion section.

Line 293 - "which makes very important": which makes it important.

Line 313 – "should be also taken into account": should also be taken into account.

Line 320 – "many field": many fields.

Figure 2: The most important components of this figure, the radar locations and their 100-km range, are relative hard to distinguish from the rest of the map with the using colors. My suggestion would be to use different colors and somewhat larger font and icon size.

Figure 8: The figure has a title and a footnote, which are partially repeated in the caption. I would suggest to just keep the text in the caption and remove the title and footnote from the figure. In addition, this holds for some other figures, too.

Figure A2: What is the highest value in the sub panels? It seems that the maximum value of the color scale could be somewhat lower, e.g. 50 mm, that would give somewhat more contrast.

**References**

Goudenhoofdt, E. and Delobbe, L.: Generation and Verification of Rainfall Estimates from 10-Yr Volumetric Weather Radar Measurements, Journal of Hydrometeorology, 17, 1223–1242, https://doi.org/10.1175/jhm-d-15-0166.1, 2016.

---

## Author Comment (AC1)

Dear Ruben Imhoff,

Thank you very much for the very careful reading of the manuscript and the detailed evaluation. Your constructive comments and suggestions were greatly appreciated. Please find below the list of your points (italicized) together with the answers (in normal font). We also provide the marked manuscript that highlights all modifications versus the first submission. We sincerely hope that the revisions meet your expectations.

Kind regards,

Michel Journée
Edouard Goudenhoofdt
Stéphane Vannitsem
Laurent Delobbe

**General comments**

*General comment 1: Rainfall product description*
The description of the method has been improved. It is important to note that the methods cannot be described in full details due to their complexity. However, a new appendix has been added in the revised version where radar experts will find more information on the significant improvements regarding the treatment of static non-meteorological echoes.

*General comment 2: Description of the methods*
We carefully considered the possibility to move all the methods in a dedicated section. We think, however, that our paper does not fit well on the typical "Data-Methods-Results" structure as we do not really propose any methodological novelty. Regarding the data analysis section (Section 3), most methodological descriptions are furthermore short (except for the NMF, but the background information about NMF is now moved to the Appendix) and most methods are standard/generic (e.g., inverse distance weighted interpolation, cross validation). We therefore prefer to keep the methodological descriptions close to the discussion of their results. We are also convinced that going to a typical "Data-Methods-Results" structure will impair the readability of our manuscript.

*General comment 3: Application of areal averages (section 3.4)*
The Section 3.4 has been complemented with an analysis to illustrate how the difference between the IDW of rain gauge data and the radar product evolves when they are aggregated in space and time. For this, we derived precipitation values for 67 catchments of various sizes and for accumulation durations from 1 to 72 hours based on the IDW and RADFLOOD21 datasets. Figure 14 shows in terms of MAE how the discrepancies between both datasets decrease when the rainfall data are aggregated in space over larger domains and accumulated over longer periods.
In addition, a figure with the cumulative precipitation sums derived from the timeseries in Figure 12 has been added (see Figure D4).
Regarding the suggestion to apply an hydrological model for simulating discharges and analyzing the results making use of the two rainfall datasets, we agree that it would be a very

interesting extension of the present study. However, we consider that it is outside the scope of this paper which is really focused on documenting rainfall. Furthermore, such additional study is clearly outside our field of expertise.

**Specific comments**

*Lines 27 – 29: What does this exactly mean for hydrology and/or the flooding that took place?* These measurements bring additional information on the event and its dynamics and complexity. The following sentences have been added in the introduction of the revised version: "The increase of the seismic noise during the event is induced by the rising stream turbulence, sediment and debris transport. On 15 July at 00:15 UTC a sudden increase of the seismic noise coincides with a detailed testimony reporting a sudden roaring in the valley before the arrival of a flash flood, described as a "tsunami", 3 km downstream of the geophysical station. The gravimeter is installed 48m underneath the surface and signal variations are induced by water accumulating above the gravimeter. The evolution of the gravity measurements along the event shows increasing subsoil's saturation with less and less water accumulation and increasing runoff."

*Lines 70 – 74: What was the density (per km2) of the gauging network and how does this density differ per region. I.e., how is the density in a heavily hit catchment such as the Vesdre, for instance? It's sometimes hard to estimate that from the figure.* A map of Belgium with the definition of the discussed catchments and the locations of the rain gauges has been added in the Introduction. In complement, the following sentences have been added in Section 2.1: "The spatial distribution of these 308 sites is illustrated in Figure 1 by the grey dots. On average over Belgium, this represents a density of 10 measurement sites per 1000 km. This density is somewhat lower for the Meuse basin (9 per 1000 km) and slightly larger for the Vesdre basin (11.5 per 1000 km)."

*Lines 81 – 82: How did this gap filling exactly work? Were gaps replaced with the data of the closest station, was some kind of spatial interpolation method used or something else?* Weather observations are routinely checked at RMI against errors. This data quality control (QC) task is performed manually with the help of data interfaces that include statistical plausibility tests, time series plots, maps, etc. In the case of missing rain gauge data, several estimations are suggested by the QC interface: data from the nearest rain gauge, inverse distance weighted interpolation (IDW) from neighboring rain gauges, radar-based QPE, etc. In the present case, the estimations were derived by inverse distance weighted interpolation (IDW) from neighboring rain gauges. This is now specified in the revised manuscript: "Estimations derived by inverse distance weighted interpolation (IDW) from neighboring weighing rain gauge data have been considered to correct the 2 periods of erroneous data as well as to fill the gaps."

*Lines 82 - 84: How were the daily values adjusted? This, as well as my previous comment, is quite relevant information for the dataset you are providing.* If a manual observation is made later than 08:00 local time and if it is raining between 08:00 and the actual observation time, then the daily accumulation of the preceding day will be

overestimated while the accumulation of the following day will be underestimated. When such a case is detected, the daily accumulations are adjusted to follow the evolution of the daily totals of neighboring rain gauges. The following sentence has been added in Section 2.1 "This adjustment is based on the sequence of daily totals of neighboring rain gauges."

Globally, the data quality control implied very few interventions on the data, and none for the rain gauges located in the most severely affected area (e.g., Vesdre Basin). This is now stated in the revised manuscript:

"Globally, the QC analysis led to few interventions on all available rain gauge data and none concerning the rain gauges located in the most severely affected areas (i.e., Vesdre basin)."

*Line 104 – "different kind of technology": do you mean that the hardware of the radars is different?*

The radars are all C-band dual-polarization weather radars except the Wideumont radar which was still a single-polarization radar in 2021. Next to this difference, the radars also differ by their configuration and data processing. This has been clarified in section 2.2.1 of the revised manuscript.

*Lines 104 – 105: This is indeed a major challenge. It would be interesting for the reader if the authors can briefly describe how RMI handles this to make their composite.*

This is complex and technical. The main approach to reduce inhomogeneities between radars is the use of a static clutter map and a long-term gauge bias correction for each radar. These two steps are described briefly in sections 2.2.2 and 2.2.4 and in a new appendix.

*Lines 109 – 112: Could the authors describe in a little more detail what this basic radar estimation method is (just the Marshall-Palmer reflectivity to rain rate transformation, or more?) and how the comparison with the rain gauges took place. Regarding the latter, we often compare gauge location by corresponding grid cell, but it is also possible to take the rainfall advection into account (as the radar and rain gauges measure at different elevations), etc.*

The description has been extended in section 2.2.2. The advection error is not taken into account. This error is relatively random and can be mitigated by taking median values.

*Lines 113 – 120: I think this is a major improvement of the radar rainfall estimation in hilly regions. Actually, this improvement makes the provided and described dataset unique and different from what was already available at the time. Although I am familiar with the method used, I do not think that the average reader is. Hence, can I ask the authors to put more emphasis on this procedure and describe it in more detail.*

In the present case, this improvement is indeed very important for producing high-quality rainfall estimates at ground. The importance of the method is already stressed in section 2.2.2 of the manuscript. The method is now described with more details in the appendix.

*Lines 123 – 124: What was considered abnormal or unrealistic, i.e. was there a threshold or is this qualitative?*

The clutter removal techniques are extensively described in Goudenhoofdt and Delobbe (2016). Concerning the identification based on the vertical profile of reflectivity, the technique is based on thresholds and it is described in the reference paper as follows : "A measurement at a given elevation is considered as clutter if the gradient between its value and the corresponding (horizontally) interpolated value on a higher (lower) elevation exceeds in magnitude -20 dBZ km$^{-1}$ (+10 dBZ km$^{-1}$). Because of variations from signal fluctuations, a minimum absolute difference of 5 dBZ between two corresponding values at different elevations is required for clutter identification".

*Lines 129 – 130: How is this identification of the precipitation type exactly done and what Z-R relationship is used? I know most of this is described in Goudenhoofdt and Delobbe (2016), but either refer explicitly to this paper or (/and) describe it here.*
Some modifications have been applied since the publication in 2016. The relations for rain, relevant for this event, are now explicitly described in section 2.2.3.

*Lines 136 – 137 - "The rainfall accumulation over 5 minutes is obtained by computing the movement of precipitation using optical flow techniques": What kind of optical flow techniques were used?*
The combined local-global method is used. Note that the flood case was not characterized by strong winds. Temporal sampling effects (e.g. jumping cell or ripple effects on the accumulation maps) were therefore very limited. This information has been added in section 2.2.4.

*Lines 139 – 143: If any methods from other papers were used in the KED method and setup, then some referencing is necessary here.*
Although no specific external references have been used, Hudson and Wackernagel (1994) is now cited for background information about KED. More details are also provided in appendix B.

*Lines 160 – 161: Although I fully agree, what can be said about the quality of the radar here? I.e., is the radar missing any precipitation in space, or overestimating, in this region? As radar estimates in hilly and mountainous areas are challenging, this is to be expected. It would be very interesting to have an idea of this too. This could also point out why and where the combination of both data sources is going to be crucial for a good analysis of this event. The answer to this comes already partially back in Figure 4, so good to give some more details here.*
It is difficult to assess the quality of the radar product regarding the spatial structure of the most extreme precipitation because the density of rain gauges is rather low with respect to the size of the area affected by the most extreme 3-day accumulations (e.g., above 200mm). We nevertheless complemented the discussion about uncertainty/accuracy with the following sentences:
"By taking a closer look on the rightmost point of the scatter plots of Figure 5 (i.e., manual rain gauge with the largest 3-day accumulation), we find an underestimation of almost 25% by the radar product, while the IDW interpolation of neighboring rain gauges underestimates that specific observation by 35%. A similar analysis but focused on daily rainfall totals also highlights that the estimation of the largest values is challenging but better addressed by the radar product. Daily precipitation values above 80mm, which have been recorded 17 times by manual rain gauges during the 3-day period, are on average underestimated by 16% by the radar product and by 22% by the IDW interpolation of neighboring rain gauges."
A paragraph about uncertainties in rain gauge data has also been added at the end of Section 2.1.

*Lines 164 – 168: This actually belongs in the methods section, which is right now missing for the analysis part of this paper.*
For the sake of readability of the paper, we prefer not to move these lines (see our answer to the 2nd general comment).

*Lines 175 – 176: Could the authors briefly describe the used method (and reason to use that method) from Van de Vyver (2012, 2013) in the methods section and then refer to the paper?*
The following sentences have been added to briefly describe the considered extreme precipitation statistics:
"In order to assess the exceptionality of the precipitation amounts during the event, extreme precipitation statistics derived for Belgium by Van de Vyver (2012) and Van de Vyver (2013) have been considered. These statistics result from a spatial generalized extreme value (GEV) distribution fitted on annual rainfall maxima derived for various accumulation durations and several locations in Belgium from historical precipitation data. Thanks to this spatial approach, extreme precipitation statistics are estimated for any location in Belgium and are currently considered as the reference for, e.g., the design of hydraulic structures. Return periods can be associated from these statistics to the 3-day precipitation accumulations, as illustrated in Figure 6 for the rainfall distribution estimated by the radar product (i.e., Figure 4b)."

*Lines 206 – 208 – "For longer accumulation durations (from 6 hours to 3 days), a clear trend towards extreme values can be noticed with, in parallel, an increase in the size of the severely affected areas.": I think the authors can put some more emphasis on this finding. It would be interesting to relate this to literature findings (either here on in a discussion section) for similar extreme flooding events or similar rainfall patterns. I.e., is this what you expect or is this unique to this event?*
This finding illustrates the highly unusual nature of this event. In Belgium, extreme summer rainfalls are generally caused by local convective storms and, consequently, extreme return periods are obtained for short durations and concern relatively small areas. In surrounding regions with similar climatic conditions, this type of event is also very rare. A relatively similar event with mixed convective and stratiform rainfall occurred in the Netherlands and the bordering part of Germany in August 2021 (Bauer et al., 2011). Over an area of 740 km2 more than 120 mm of rainfall were recorded in 24 h. Maximum amounts for the whole episode reached 160 mm, which is still lower than the extreme values obtained in the present case.
Long duration rainfall events, affecting large areas and producing flash floods in several catchments are relatively common in the Mediterranean and Alpine-Mediterranean regions as shown in several inventories of European flood events (Gaume et al., 2009; Marchi et al., 2010 and Amponsah et al., 2018). These events occcur mostly in Autumn under specific climatic forcing and triggering mechanisms that are not present in the climatic region of interest here.
The above comments have been included in Section 3.2 near Fig. 8. in the revised version.

*Lines 221 – 243: This belongs in the methods section.*
The background information about NMF was moved to the Appendix C.

*Section 3.3: I am not familiar with the NMF method, but can I ask the authors to explain a*

*bit more elaborately what you are trying to reach with this method and why you focus in the results on a rank-3 NMF (and not more or less)? Why do we need the NMF method? Is it to be able to split the mesoscale rainfall field into more regional scale stratiform and convective rainfall, or is there another reason? See also my comment below related to figure 10a.*

Our objective with the NMF analysis is to illustrate that the temporal distribution of rainfall was not everywhere the same. NMF decompositions of various ranks have been derived, but the rank-3 NMF analysis provides the clearest decomposition of the data into distinct spatio-temporal patterns as shown in Figure 10. Figure 10 provides a very synthetic view of the event with 3 maps and 3 timeseries. It allows to derive the following qualitative descriptions of the event:

> In the south-east part of the country, rainfall were rather continuous on 13th and 14th July. 15th July as rather dry in that area.

> In the central part of the country, few rainfall were observed on 14th July, although it was the most severe period for the Vesdre basin. For the central part of the country, rainfall responsible for floods occurred on 15th July.

> In the East part of the country (e.g., Vesdre basin), we can note a sequence of very intense precipitation peaks from the evening of 13th July to the morning of 15th July.

We find it interesting to highlight these very different spatio-temporal rainfall patterns, even if we do not have yet determined what the dynamical source of these patterns is.

The following sentences have been added in the revised manuscript (see Section 3.3):

"This NMF analysis will allow us to highlight that rainfall was distributed in time differently over the country. The NMF analysis will provide a synthetic view of the event from which qualitative descriptions can be derived."

"Although NMF approximations of various ranks have been derived, the discussion will be focused on the rank-3 NMF that provides a clear decomposition of the data into 3 distinct spatio-temporal patterns as illustrated in Figure 10."

*Figure 3: Here I would show the catchments that you are focusing on instead of the province of Liège.*

The Province of Liège is shown on Figure 4 just to highlight the area where the most extreme precipitation occurred. It allows mainly to link to zoomed maps (bottom line) to the maps of Belgium (top line).

*Figure 10a: The sub panel is showing the difference between the rank-3 NMF and RAD-FLOOD21 data, right? This is not yet clear from the caption. In addition, from the figure and the corresponding text it is not directly clear to me if a difference is good or bad and how big of a difference is considered acceptable. As mentioned earlier, I am not familiar with the NMF method, so that definitely plays a role too. Is a small difference and indication that with (just) three ranks, and corresponding regions, you capture most of the rainfall in space and time, or should I interpret it differently?*

The caption of Figure 10a has been clarified. Since NMF provides an approximation of the initial data, there is a difference on the 3-day total between the rank-3 NMF and RADFLOOD21 data, as shown on Figure 11a. The map on Figure 11a indicates where the discrepancies between both is the largest. In particular, there is a zone in central Belgium where the rank-3

NMF approximation underestimates the 3-day total by up to 50mm. This is a pattern not captured by the rank-3 NMF, but where total rainfall amounts remain rather moderate (i.e., lower than 100mm over the 3 days).

*Figure 12: Useful figure, but I think this figure fits better in the study area description. Hence, put it together with one of the first figures. I think it would fit well in figure 3.*
The map of Belgium with the definition of the discussed catchments has been moved to the Introduction section.

*Figure 13: It looks like the individual points in the graphs have relatively coarse steps (1, 2, 3, 6, 12, etc. hours), even though the data is based on hourly accumulations. Hence, I think the authors could put more data points in this graph, which more clearly identifies when differences between the two datasets occur. In fact, why not just go to 5-min steps and derive the cumulative sums on that time step?*
The Figure 13 in the revised manuscript illustrates now maximum precipitation accumulations derived for durations from 1 to 72 hours by steps on 1 hour. Regarding the use of the hourly radar product instead of the 5-min one, we consider the analysis based on hourly data as meaningful since the event starts to be extreme for accumulation durations larger than 6 or even 12 hours (see Figures 8 and 9).

*Figure A1: Just to double check, we are looking at the maximum x-hour window accumulation that occurred, right? So, for e.g. the grids cells in the Vesdre, we are looking at the statistics of the maximum x-hr accumulation that occurred during the 3-day period and not the average? Besides, do you look at a rolling window in which for every 5-min steps a new x-hour window is assessed or have the authors assessed it differently? Good to briefly mention this in the methods and the caption.*
In Figure A1 (now Figure D1), we indeed consider for each grid cell the maximum x-hr accumulation that occurred during the 3-day period. This maximum x-hr accumulation is derived from the hourly radar product for durations till 12 hours, and otherwise from daily-aggregated radar product. This seems to us already clearly mentioned in the text and caption:
Text, Section 3.2: "The analysis of Figure 8 can be extended to any location in Belgium based on the **hourly** radar product. For each 1-km pixel of the radar product, the **maximum rainfall accumulation is derived for durations from 1 hour to 3 days during the period** July 13th 06:00 UTC to July 16th 06:00 UTC."
Caption of Fig. A1 (now D1): "Return period estimated for the **maximum** precipitation accumulation between 13th July 06:00 UTC to 16th July 2021 06:00 UTC for durations from 1 hour to 3 days. The **maximum accumulations are derived from hourly (resp. daily) radar product data for durations till 12 hours (resp. of 1, 2 or 3 days)**."

**Technical corrections**

*Line 107: Although true, I think this needs some referencing.*
A reference has been added.

*Lines 130 – 131: Good to add a reference here.*
The Z-R relationship is mentioned here `https://www.ou.edu/radar/z_r_relationships`.

pdf (see page 2) with a reference to `http://www.roc.noaa.gov/ops/z2r_osf5.asp` which is not anymore active. No public reference from NOAA seems currently available.

*Lines 275 – 278: From a hydrologist's perspective, it is more useful to provide these numbers in mm instead of in km3. The same comment holds for the conclusion section.*
Total precipitation values in km3 were requested by the Editor. Values in mm have been added between brackets.

*Figure 2: The most important components of this figure, the radar locations and their 100-km range, are relative hard to distinguish from the rest of the map with the using colors. My suggestion would be to use different colors and somewhat larger font and icon size.*
The figure has been modified accordingly.

*Figure 8: The figure has a title and a footnote, which are partially repeated in the caption. I would suggest to just keep the text in the caption and remove the title and footnote from the figure. In addition, this holds for some other figures, too.*
Most figures had internal titles and sometimes subtitles and footnotes in order to be comprehensible in themselves (e.g., in case they are reused outside the manuscript). We removed internal titles from several figures (Figures 6, 7, 8 and 9 in the revised manuscript). For other more complex figures, we think that summarized internal titles are still useful, in addition to a well-detailed caption.

*Figure A2: What is the highest value in the sub panels? It seems that the maximum value of the color scale could be somewhat lower, e.g. 50 mm, that would give somewhat more contrast.*
The largest 3-hourly values reach almost 55mm (on 2021-07-14 from 18:00 to 21:00 UTC). The colorscale has been adapted to give slightly more contrast.

All language-related suggestions have been implemented in the revised manuscript.

**References**

[revised manuscript text omitted]

---

## Author Comment (AC2)

Dear Wolfgang Wagner,

Thank you very much for your evaluation of the manuscript. Please find below the list of your points (italicized) together with the answers (in normal font). We also provide the marked manuscript that highlights all modifications versus the first submission. We sincerely hope that the revisions meet your expectations.

Kind regards,

Michel Journée
Edouard Goudenhoofdt
Stéphane Vannitsem
Laurent Delobbe

*I do not have much to add to the excellent review by Ruben Imhoff, except to say that personally I am not sure about the value of the principal component analysis. Rather than interpreting the complex spatiotemporal patterns in Figure 9, I would much prefer to see a catchment based analysis of the rainfall dynamics to get a better grasp of the hydrological processes (as also pointed out by Ruben Imhoff).*

Our objective with the NMF analysis is to illustrate that the temporal distribution of rainfall was not everywhere the same. NMF decompositions of various ranks have been derived, but the rank-3 NMF analysis provides the clearest decomposition of the data into distinct spatio-temporal patterns as shown in Figure 10. Figure 10 provides a very synthetic view of the event with 3 maps and 3 timeseries. It allows to derive the following qualitative descriptions of the event:

> In the south-east part of the country, rainfall were rather continuous on 13th and 14th July. 15th July as rather dry in that area.

> In the central part of the country, few rainfall were observed on 14th July, although it was the most severe period for the Vesdre basin. For the central part of the country, rainfall that causes floods occured in 15th July.

> In the East part of the country (e.g., Vesdre basin), we can note a sequence of very intense precipitation peaks from the evening of 13th July to the morning of 15th July.

We find it interesting to highlight these very different spatio-temporal rainfall patterns, even if we do not have yet determined what the dynamical source of these patterns is.

The following sentences have been added in the revised manuscript (see Section 3.3):

"This NMF analysis will allow us to highlight that rainfall was distributed in time differently over the country. The NMF analysis will provide a synthetic view of the event from which qualitative descriptions can be derived."

"Although NMF approximations of various ranks have been derived, the discussion will be focused on the rank-3 NMF that provides a clear decomposition of the data into 3 distinct spatio-temporal patterns illustrated in Figure 10."

*Maybe the authors could also add a comparison of rainfall and runoff time series for selected catchments, just to see what the observed rainfall amounts meant on the ground.*

We agree that it would be interesting to present such results. However, we consider that it is outside the scope of this paper which is really focused on documenting rainfall. Presenting the hydrological impact is also outside our field of expertise.

[revised manuscript text omitted]

---

## Referee Report (RR1)

Review of: **Quantitative rainfall analysis of the 2021 mid-July flood event in Belgium** by Michel Journée et al.

Ruben Imhoff

Ruben.Imhoff@deltares.nl

July 4, 2023

**Summary**

I would like to thank the authors for the work they have put in the revisions. I think the manuscript has significantly improved and I am quite satisfied with the changes as made by the authors, based on the suggestions from the reviewers. Especially the emphasis on the new methods and the addition of extra analyses within the areal averages section make the work seem stronger.

Regarding the structure of the manuscript with methods sometimes throughout the results section, I understand the point of view of the authors, and although it is not my preference for readability, I think it works as is.

Overall, I have only a few minor comments left at this point and I am looking forward to seeing the work in its published form.

**Minor comments (in blue, italic are the responses from the authors)**

*Rainfall product description:* I would like to thank the authors for the extensive description they have added to the appendix. I am aware that the full method is too long to describe, but the proposed additions are more than sufficient, thanks. As a final remark regarding this point, can I ask the authors to clearly emphasize in the appendix the addition described in lines 136 – 137 "*A new method has been developed to keep values as actual precipitation if they are significantly larger than the maximum expected clutter level.*"? This is just a minor remark, but good to point out the significance of this change for the dataset in the appendix as well.

Lines 123 – 124: What was considered abnormal or unrealistic, i.e. was there a threshold or is this qualitative? "*The clutter removal techniques are extensively described in Goudenhoofdt and Delobbe (2016). Concerning the identification based on the vertical profile of reflectivity, the technique is based on thresholds and it is described in the reference paper as follows : "A measurement at a given elevation is considered as clutter if the gradient between its value and the corresponding (horizontally) interpolated value on a higher (lower) elevation exceeds in magnitude -20 dBZ km−1 (+10 dBZ km−1). Because of variations from signal fluctuations, a minimum absolute difference of 5 dBZ between two corresponding values at different elevations is required for clutter identification".*

Although I am happy with the answer, I cannot find it back it the adjusted text. If I did not miss it by accident, could the authors still add it?

Figure 10a: The sub panel is showing the difference between the rank-3 NMF and RADFLOOD21 data, right? This is not yet clear from the caption. In addition, from the figure and the corresponding text it is not directly clear to me if a difference is good or bad and how big of a difference is

considered acceptable. As mentioned earlier, I am not familiar with the NMF method, so that definitely plays a role too. Is a small difference and indication that with (just) three ranks, and corresponding regions, you capture most of the rainfall in space and time, or should I interpret it differently? "*The caption of Figure 10a has been clarified. Since NMF provides an approximation of the initial data, there is a difference on the 3-day total between the rank-3 NMF and RADFLOOD21 data, as shown on Figure 11a. The map on Figure 11a indicates where the discrepancies between both is the largest. In particular, there is a zone in central Belgium where the rank-3NMF approximation underestimates the 3-day total by up to 50mm. This is a pattern not captured by the rank-3 NMF, but where total rainfall amounts remain rather moderate (i.e., lower than 100mm over the 3 days).*"

Could the authors also add the last sentence "but where total rainfall amounts remain rather moderate" of their explanation to the main text? I think this will be insightful for the readers.

Figure 14 and A2: Thanks for adding both figures! The used rainbow color map may not be the best choice for colorblind people. The authors may consider a different color scheme, see e.g. Crameri et al. (2020).

**References**

Crameri, F., Shephard, G.E. & Heron, P.J. The misuse of colour in science communication. Nature Communications 11, 5444 (2020). https://doi.org/10.1038/s41467-020-19160-7

---

## Author Response (AR2)

Dear Ryan Teuling, dear Editor,

We thank you for the rapid peer-reviewing of our revised manuscript.
We have addressed the last points raised by Ruben Imhoff (see details below as well as the marked manuscript).
We sincerely hope that the final manuscript meets your expectations for publication in HESS.

Kind regards,

Michel Journée
Edouard Goudenhoofdt
Stéphane Vannitsem
Laurent Delobbe

**Responses to the comments of Ruben Imhoff:**

(The comments from the reviewer are in italic and our responses are in normal font.)

*Rainfall product description: [...] This is just a minor remark, but good to point out the significance of this change for the dataset in the appendix as well.*
This is now also clarified in the Appendix A4.

*Lines 123 – 124: [...] Although I am happy with the answer, I cannot find it back it the adjusted text. If I did not miss it by accident, could the authors still add it?*
These details are now well provided in the new Appendix B.

*Figure 10a: [...] Could the authors also add the last sentence "but where total rainfall amounts remain rather moderate" of their explanation to the main text? I think this will be insightful for the readers.*
"but where total rainfall amounts remain rather moderate" is now in the main text (lines 299 and 300).

*Figure 14 and A2: Thanks for adding both figures! The used rainbow color map may not be the best choice for colorblind people. The authors may consider a different color scheme, see e.g. Crameri et al. (2020)*
The color scales have been changed.